# Behavior-Invariant Task Representation Learning with Transformer-based World Models for Offline Meta-Reinforcement Learning

**Fuyuan Qian**[* 1]   **Menglong Zhang**[* 2]   **Song Wang**[1]   **Quanying Liu**[1 3 4]

## Abstract

Offline meta-reinforcement learning leverages static datasets to enable agents to generalize to unseen environments by combining offline efficiency with meta-learning adaptability, yet it faces key challenges from context and policy distribution shifts. These issues hinder agents from adapting to online environments, and are further exacerbated under sparse-reward settings. As a result, agents often become trapped in an inherent pattern dilemma, failing to achieve robust generalization. In this work, we propose a novel framework that integrates information-theoretic task representation learning with a Transformer-based stochastic world model. Our approach extracts task-defining latent variables that are invariant to behavior policy, thereby effectively mitigating the context distribution shift. To further handle policy shift and model exploitation, we apply a conservative value penalty to imagination-based rollouts, preventing the policy from exploiting model inaccuracies while maintaining robust adaptation. Extensive evaluations demonstrate that our method outperforms state-of-the-art approaches, with superior stability and generalization under out-of-distribution and sparse-reward settings.

## 1. Introduction

In meta-reinforcement learning, agents adapt to unknown environments by learning representations from similar tasks and exploiting prior knowledge (Finn et al., 2017; Duan

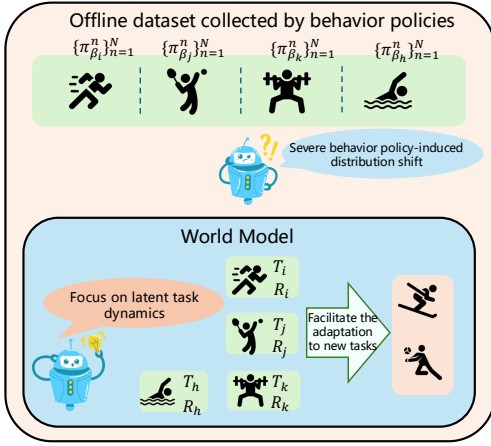

*Figure 1.* Modeling latent task dynamics with a world model facilitates the capture of behavior-invariant task information, thereby supporting task inference and enabling rapid adaptation.

et al., 2016; Rakelly et al., 2019; Zintgraf et al., 2020). Compared to online meta-RL, offline meta-RL is required to model a static task distribution and ensure that policies learned from offline data can robustly transfer to unseen tasks and subsequent online deployment while maintaining strong generalization (Dorfman et al., 2021; Li et al., 2020; Gao et al., 2023; Li et al., 2024). When reward signals are sparse, effective adaptation increasingly depends on learning robust representations of task dynamics and implicit transfer relationships to handle distributional shifts (Packer et al., 2021; Grigsby et al., 2024; Zhang et al., 2025b). Nevertheless, current offline meta-RL approaches struggle to learn such representations in sparse-reward and out-of-distribution environments.

Moreover, offline meta-RL faces two coupled distribution-shift challenges. First, context distribution shift arises because the contexts used for task inference during meta-training are sampled from static datasets collected by task-dependent behavior policies, whereas during meta-testing the agent must infer the task from contexts collected online by its exploration policy. The mismatch between behavior and exploration policies can make the learned context encoder unreliable on test-time contexts, leading to degraded task inference and generalization (Yuan & Lu, 2022; Gao et al., 2023; Ajay et al., 2022). Second, policy distribu-

---

[*]Equal contribution  [1]Department of Biomedical Engineering, Southern University of Science and Technology, Shenzhen, Guangdong, China [2]AI Thrust, The Hong Kong University of Science and Technology (Guangzhou), Guangzhou, Guangdong, China [3]Omni-Intelligence, Shenzhen, Guangdong, China [4]Shenzhen Loop Area Institute, Shenzhen, Guangdong, China. Correspondence to: Quanying Liu <liuqy@sustech.edu.cn>, Menglong Zhang <mzhang943@connect.hkust-gz.edu.cn>.

*Proceedings of the 43rd International Conference on Machine Learning*, Seoul, South Korea. PMLR 306, 2026. Copyright 2026 by the author(s).

tion shift arises because offline datasets cannot cover the full state-action space. When the learned policy selects out-of-distribution actions during optimization, the critic may assign overestimated values to unsupported regions, a problem that becomes more complex in offline meta-RL due to the need for cross-task generalization (Mitchell et al., 2021; Wang et al., 2023). Together with MDP ambiguity (Dorfman et al., 2021), these issues give rise to an *inherent pattern dilemma*, where agents overfit to behavior-policy-specific patterns encoded in the offline data, limiting their ability to generalize efficiently to unseen environments.

World models (Ha & Schmidhuber, 2018; Hafner et al., 2020) provide a natural mechanism for learning predictive dynamics and performing planning through latent imagination. By compressing observations into a compact latent space and predicting how latent states evolve under actions, a world model allows an agent to evaluate potential future outcomes without directly interacting with the real environment. This ability has led to strong performance in complex video games (Hafner et al., 2021; 2025a;b), robotic control (Wu et al., 2023), and autonomous driving (Wang et al., 2024a;b). By modeling task-specific latent dynamics in a compact latent space, world models help avoid overfitting to the empirical task distribution and partially filter out the effects induced by heterogeneous behavior policies. This property naturally aligns with the requirements of offline meta-RL for learning robust task representations (Figure 1).

In this work, we propose **MetaSTAR**, a novel offline **Meta**-reinforcement learning framework with a **S**tochastic **T**ransformer-based world model for beh**A**vior-invariant task **R**epresentations. Motivated by the information-theoretic view of causal disentanglement, MetaSTAR constructs a stochastic world model to predict forward latent dynamics in a compact state space. By forcing the model to capture transition-related and reward-related evolution rather than behavior-policy-specific correlations, MetaSTAR learns dynamics-centric task representations that are less sensitive to heterogeneous data-collection policies, thereby mitigating context distribution shift. To further address policy distribution shift in offline meta-RL, MetaSTAR combines the learned world model with contextual imagination and conservative policy optimization. The world model generates context-conditioned imagined rollouts, while the conservative value objective penalizes unsupported state-action regions and discourages the policy from exploiting model errors. This design enables safer latent-space imagination and more reliable transfer from offline training to online adaptation, especially under sparse rewards and out-of-distribution tasks. Our main contributions are as follows:

- From an information-theoretic perspective on task representation, we propose MetaSTAR that leverages world models to learn behavior-invariant representa-

tions in offline meta-reinforcement learning.
- We theoretically validate the effectiveness of world models in filtering the impact of behavior-policy-induced distribution shift during the meta-training stage, providing principled justification for behavior-invariant task representation learning.
- Empirically, MetaSTAR alleviates the inherent pattern dilemma and achieves strong online adaptation performance across a wide range of sparse-reward and out-of-distribution tasks.

## 2. Preliminaries

### 2.1. Problem Formulation

We consider a context-based offline meta-reinforcement learning setting, where tasks are drawn from a distribution $p(\mathcal{M})$. Each task $M_i \in \mathcal{M}$ is modeled as a Markov Decision Process (MDP) defined by the tuple $M_i = (\mathcal{S}, \mathcal{A}, T_i, R_i, \rho_0, \gamma)$. All tasks share the same state space $\mathcal{S}$ and action space $\mathcal{A}$, while differing in their transition dynamics $T_i$ and reward functions $R_i$. In the offline setting, the agent has access only to a static dataset $\mathcal{D} = \{\mathcal{D}_i\}_{i=1}^{N_{\text{tasks}}}$, where each $\mathcal{D}_i$ consists of transitions $\{(s_j^i, a_j^i, r_j^i, s_j'^i)\}_{j=1}^{N_{\text{size}}}$ collected by its corresponding behavior policies $\{\pi_{\beta_i}^n\}_{n=1}^{N_{\text{update}}}$. Given a context $c_i \subset \mathcal{D}_i$, the agent first infers a task representation $z_i$ via an encoder $q_\phi(z_i \mid c_i)$. The learned task representation $z_i$ is then concatenated with the state $s$ and provided as input to the policy $\pi_\theta(a \mid s, z_i)$ for policy learning. The overall optimization objective can be expressed as:

$$\max_{\theta, \phi} \ \mathbb{E}_{M_i \sim p(\mathcal{M}), \, c_i \sim \mathcal{D}_i, \, z_i \sim q_\phi(z|c_i)} \Big[ J_{M_i}\big(\pi_\theta(\cdot \mid s, z_i)\big) \Big]. \tag{1}$$

The agent is required to infer task-specific dynamics from limited context and adapt to unseen tasks. As discussed in the introduction, offline meta-RL is challenged by both context distribution shift and policy distribution shift, which can make task inference unreliable and policy improvement unstable. These issues are further amplified under sparse rewards and out-of-distribution task adaptation, where task-identifying signals are limited and offline datasets provide incomplete coverage of the state-action space. The above issues can be summarized as follows: the learned patterns are overly rigid, resulting in limited exploratory behavior and driving agents into an *inherent pattern dilemma*. Since offline datasets are inevitably shaped by the behavior policies used for data collection, task representations learned without appropriate constraints may capture correlations that are predictive within the training distribution but fail to transfer to unseen tasks. Consequently, it is crucial to learn **behavior-invariant task representations** (Definition 2.1) that preserve task-relevant transition and reward structures

while suppressing behavior-policy-specific variations. Such representations are essential for handling task distribution shifts and achieving robust domain adaptation.

## 2.2. Information-Theoretic Task Representation

To formally define behavior-invariant task representations from an information-theoretic perspective, we adopt the framework proposed in UNICORN (Li et al., 2024) which posits that an ideal task representation $Z$ should maximize the mutual information $I(Z; M)$ with the task variable $M$. However, since the true task identity $M$ is unobservable in practice, the representation can only be inferred from the observed context sequence $X$. Following the causal decomposition, the context variable $X$ is decoupled into behavior-related components $X_b = (s, a)$ and task-related components $X_t = (r, s')$. By the chain rule, this yields

$$I(Z; X) = \underbrace{I(Z; X_t \mid X_b)}_{\text{Primary Causality}} + \underbrace{I(Z; X_b)}_{\text{Lesser Causality}} . \qquad (2)$$

Based on this decomposition, UNICORN introduces the following central theorem to bound $I(Z; M)$:

$$I(Z; X_t \mid X_b) \le I(Z; M) \le I(Z; X_t \mid X_b) + I(Z; X_b). \qquad (3)$$

Therefore, robustness to context shift can be promoted by maximizing the *primary causality* $I(Z; X_t \mid X_b)$, while suppressing the *lesser causality* $I(Z; X_b)$. Intuitively, $Z$ should preserve predictive information about future states and rewards, which characterize task dynamics, while discarding spurious correlations regarding which actions the behavior policy would inherently select.

In our work, we argue that a Transformer-based world model provides a principled realization of this objective (Theorem 3.1). By training the world model to predict future latent states from historical latent states and actions, the dynamics-learning objective encourages $Z$ to capture predictive transition-related and reward-related structures. This can be interpreted as a variational surrogate for increasing the primary causality $I(Z; X_t \mid X_b)$ in the latent space. Consequently, the learned representation is encouraged to distill behavior-invariant latent task dynamics, providing compact and informative task signals for robust task inference and adaptation.

**Definition 2.1** (Behavior-invariant task representation). Given a context sequence $X = (X_b, X_t)$ collected from task $M_i$, with $X_b = (s, a)$ and $X_t = (r, s')$, a representation $Z$ is called a *behavior-invariant task representation* if it preserves task-relevant information about $T_i$ and $R_i$ by promoting high primary causality $I(Z; X_t \mid X_b)$, while suppressing behavior-policy-specific information measured by the lesser causality $I(Z; X_b)$.

## 3. Method

In this section, we first introduce the process of modeling latent task dynamics using a Transformer-based world model (Section 3.1). We then describe how world model-based planning is combined with offline datasets to enable conservative policy learning (Section 3.2). Finally, we provide a theoretical analysis showing that optimizing the world model can be viewed as approximately learning behavior-invariant task representations (Section 3.3).

### 3.1. World Model Learning

In offline settings, data are collected by heterogeneous behavior policies, leading to diverse action distributions that can bias task representations and hinder reliable task identification. Standard context encoders may therefore capture correlations induced by behavior policies rather than task-relevant dynamics, especially under sparse rewards and out-of-distribution settings. In contrast, world models learn compact latent dynamics of the underlying MDP and naturally support long-horizon imagination. Motivated by these advantages, we employ a Transformer-based world model to learn behavior-invariant task representations.

**Latent Dynamics for Task Representation.** To learn behavior-invariant task representations, our approach models task-relevant latent dynamics rather than directly encoding trajectories in the original state space. This design reduces the influence of heterogeneous behavior policies and enables robust latent dynamics modeling across multiple offline tasks. For a trajectory $\tau^i = \{(s_t^i, a_t^i, r_t^i, s_t'^i)\}_{t=1}^T$ [1] from task $M_i$, we define the observation as $o_t = [s_t, r_{t-1}]$. Explicitly incorporating the reward from the previous time step $r_{t-1}$ as part of the observation serves as a crucial signal for characterizing task-specific MDP dynamics, and facilitates the model's ability to distinguish between different task reward functions (Zintgraf et al., 2019; Choshen & Tamar, 2023; Rimon et al., 2024). We empirically validate the importance of this design in Appendix D.6. To represent trajectories from multiple MDPs while preserving stochasticity in the latent space, we employ a variational auto-encoder (VAE) (Kingma & Welling, 2013) to map observation $o_t$ into a continuous latent distribution $E_t$. Specifically, at each time step, we sample a latent embedding $e_t \sim E_t$, which serves as the stochastic latent observation used for dynamics prediction. To capture long-horizon temporal dependencies across different tasks in offline datasets, we employ a Transformer as the deterministic component of the world model (Hafner et al., 2019; 2020; Chen et al., 2022), serving as the sequence model in the latent state space. The sequence of stochastic latent observations and actions, $\tau_{1:t}^{e,a} = \{e_1, a_1, \ldots, e_t, a_t\}$, is provided as input to

---

[1] For clarity of presentation, we omit explicit task indices when no confusion arises.

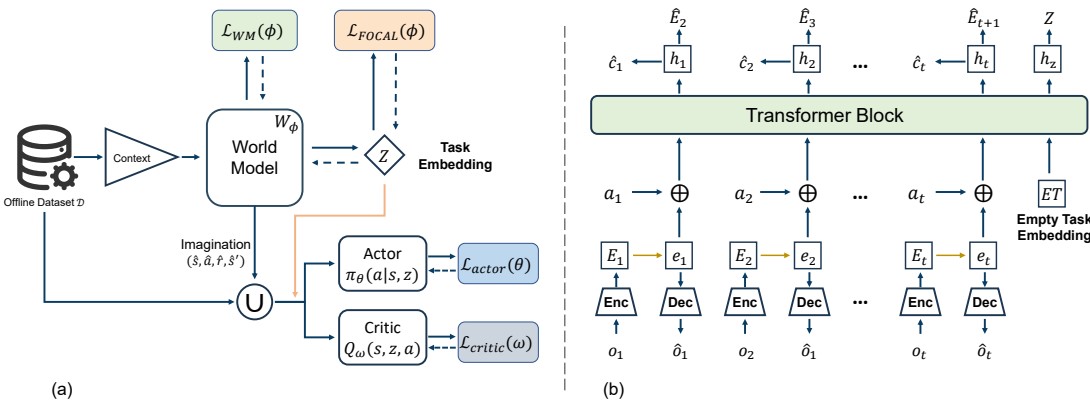

*Figure 2.* MetaSTAR Framework. (a) The world model is responsible not only for inferring task representation from context but also for augmenting the data distribution via imagination. (b) The Transformer module simultaneously handles temporal encoding and dynamic prediction, ensuring that the extracted task representations are causal.

the Transformer, which outputs a context embedding $h_t$ that aggregates historical information in the latent space. The resulting context embedding $h_t$ is then fed into two independent MLP heads: a dynamics predictor, which forecasts the latent state distribution at the next time step $\hat{E}_{t+1}$, and a continuation predictor, which predicts the continuation flag $\hat{c}_t \in \{0, 1\}$. The stochastic dynamics predictor allows the model to represent uncertainty in latent transitions, which helps reduce over-reliance on rigid patterns in offline data and improves robustness over purely deterministic representation architectures. The components of our world model are defined as follows:

$$
\begin{aligned}
\text{Observation encoder:} \quad & e_t \sim q_\phi(e_t \mid o_t), \\
\text{Observation decoder:} \quad & \hat{o}_t \sim p_\phi(\hat{o}_t \mid e_t), \\
\text{Sequence model:} \quad & h_{1:t} = f_\phi(\tau_{1:t}^{e,a}), \\
\text{Latent dynamics predictor:} \quad & \hat{e}_{t+1} \sim p_\phi(\hat{e}_{t+1} \mid h_t), \\
\text{Continuation predictor:} \quad & \hat{c}_t \sim p_\phi(\hat{c}_t \mid h_t), \\
\text{Task embedding projector:} \quad & z \sim q_\phi(z \mid h_z).
\end{aligned} \tag{4}
$$

To identify the underlying MDP from the latent history, we append a learnable query token at the end of the trajectory, as illustrated in Figure 2 (b). Through the self-attention mechanism, this embedding attends to the historical latent trajectory, aggregating task-relevant information and producing a global context embedding $h_z$. Finally, a task-specific MLP head $q_\phi(z \mid h_z)$ projects $h_z$ into the task embedding space to yield the global task embedding $z$.

**Loss Functions.** The objective is to minimize a joint loss comprising the prediction loss $\mathcal{L}_\text{pred}$, the dynamics loss $\mathcal{L}_\text{dyn}$, and the representation loss $\mathcal{L}_\text{rep}$:

$$
\mathcal{L}_\text{WM}(\phi) \doteq \mathbb{E}\left[\sum_{t=1}^{T}(\mathcal{L}_\text{pred}(\phi) + \mathcal{L}_\text{dyn}(\phi) + \beta_\text{rep}\mathcal{L}_\text{rep}(\phi))\right]. \tag{5}
$$

The prediction loss consists of two components: observation reconstruction $\log p_\phi(o_t \mid e_t)$ and continuation prediction $\log p_\phi(c_t \mid h_t)$. The former is optimized via the negative log-likelihood of $o_t$ to ensure that the stochastic latent $e_t$ preserves sufficient information for reconstruction. The latter is trained as a logistic regression task to predict episode termination. The dynamics and representation losses govern the alignment between the posterior $q_\phi(e_{t+1} \mid o_{t+1})$ and the prior $p_\phi(\hat{e}_{t+1} \mid h_t)$. We employ KL balancing with a "free-bits" threshold of 1.0 to prevent the KL divergence from dominating the total loss, thereby avoiding posterior collapse. Specifically, $\mathcal{L}_\text{dyn}(\phi)$ trains the Transformer's prior to match the encoder's posterior. By applying a stop-gradient (sg) to the encoder, we force the Transformer to internalize the underlying environment dynamics without allowing the encoder to simplify the latent space to minimize the loss. Finally, $\mathcal{L}_\text{rep}(\phi)$ ensures that observation representations remain consistent with the temporal dynamics, facilitating more stable and accurate multi-step imagination.

$$
\begin{aligned}
\mathcal{L}_\text{pred}(\phi) &\doteq -\log p_\phi(o_t \mid e_t) - \log p_\phi(c_t \mid h_t), \\
\mathcal{L}_\text{dyn}(\phi) &\doteq \max(1, \text{KL}[\text{sg}(q_\phi(e_{t+1}|o_{t+1})) \parallel p_\phi(\hat{e}_{t+1}|h_t)]), \\
\mathcal{L}_\text{rep}(\phi) &\doteq \max(1, \text{KL}[q_\phi(e_{t+1}|o_{t+1}) \parallel \text{sg}(p_\phi(\hat{e}_{t+1}|h_t))]).
\end{aligned} \tag{6}
$$

While the world model loss $\mathcal{L}_\text{WM}$ ensures that the task embedding $z$ captures the *primary causality* (Theorem 3.1), it does not explicitly guarantee that embeddings from different tasks are well-separated in the latent space. To ensure that $z$ is a discriminative and sufficient statistic for the task variable $M$, we incorporate the distance metric learning objective from FOCAL (Li et al., 2020). The FOCAL loss serves as a contrastive regularizer that optimizes the task-identifiability. This objective treats task representation learning as a distance metric learning problem, aiming to cluster embeddings of the same task while repelling those from different tasks. Specifically, given a batch of task embeddings $\{z^i, z^j\}$ ex-

tracted from different trajectory segments, the FOCAL loss is defined as:

$$\mathcal{L}_{\text{FOCAL}}(\phi) \doteq \mathbb{E}_{i,j}\Big[\mathbf{1}_{\{i=j\}}\|z^i - z^j\|_2^2$$
$$+ \mathbf{1}_{\{i \neq j\}}\frac{1}{\|z^i - z^j\|_2^2 + \epsilon}\Big]. \qquad (7)$$

By minimizing this loss, we enforce the task representation $z$ inferred from different trajectory segments of the same MDP to lie in a consistent region of the latent space, while separating embeddings from different MDPs. This regularization improves task discriminability across data collected by heterogeneous behavior policies and reduces the dependence of $z$ on behavior-related components $X_b$. The total objective for the world model in MetaSTAR is:

$$\mathcal{L}_{\text{total}}(\phi) = \mathcal{L}_{\text{WM}}(\phi) + \lambda \mathcal{L}_{\text{FOCAL}}(\phi). \qquad (8)$$

This integrated objective encourages the world model to learn latent dynamics grounded in stable task information rather than spurious variations induced by behavior policies. In particular, $\mathcal{L}_{\text{WM}}$ promotes predictive modeling of task-related components $X_t = (r, s')$, while $\mathcal{L}_{\text{FOCAL}}$ ensures that the resulting task embedding $z$ is structured and discriminative. The effect of the contrastive loss weight $\lambda$ is analyzed in Appendix D.7.

### 3.2. Agent Learning

Following the training of $W_\phi$, we proceed to learn a conservative critic $Q_\omega(s, z, a)$ and a meta-policy $\pi_\theta(a|s, z)$. The primary objective of this stage is to mitigate **policy distribution shift** by discouraging the policy from selecting unsupported state-action pairs and exploiting model inaccuracies in out-of-support regions.

**Planning with Contextual Imagination.** We use the learned world model $W_\phi$ to generate imagined rollouts for policy learning. While sequence-based world models naturally condition predictions on historical trajectories, our *contextual imagination* refers specifically to the offline meta-RL procedure of warming up the world model with real task context before imagination. This distinction is important because offline meta-RL suffers from *MDP ambiguity* (Dorfman et al., 2021): when the static context lacks sufficient identifying transitions, a meta-agent may fail to determine which MDP generated the data. Unlike standard single-task world-model rollouts, which use history mainly for predictive accuracy, contextual imagination uses real context to establish a task-consistent latent history before generating synthetic transitions for downstream policy learning. Specifically, we sample a context segment of length $L$ from the offline dataset: $\tau_{1:L} = \{(s_t, a_t, r_t, s'_t)\}_{t=1}^L$. This prefix is fed into the world model $W_\phi$ to establish a historical prior. Through its self-attention mechanism, the world

model aggregates these historical transitions to encounter potential identifying state-action pairs. Taking the terminal state $s_L$ of the warm-up sequence as the anchor, the world model continues the sequence autoregressively for a horizon of $H$: $\hat{\tau}_{L+1:L+H} = \{(\hat{s}_t, \hat{a}_t, \hat{r}_t, \hat{s}'_t)\}_{t=L+1}^{L+H}$. During this phase, actions are sampled from the current meta-policy $\hat{a}_t \sim \pi_\theta(\cdot|\hat{s}_t, z)$. By prefixing the imagination with real historical context, the generated rollouts are conditioned on evidence about the current task rather than produced from an ambiguous or uninformative latent state. This ensures that the generated transitions are not only physically plausible but also grounded in the specific MDP identified during the warm-up phase, effectively alleviating the identifiability challenges in offline meta-RL.

**Conservative Meta-Value Evaluation.** To mitigate **value overestimation** associated with policy distribution shift, we implement a conservative evaluation objective for the critic $Q_\omega(s, z, a)$. In the offline setting, the meta-policy may select actions outside the support of the offline dataset, causing the world model to generate transitions in regions where the learned dynamics are unreliable. Without regularization, the critic tends to assign erroneously high values to these regions, leading the policy to exploit model hallucinations.

Drawing on conservative model-based optimization (Yu et al., 2021), we train the critic using real transitions sampled from the offline dataset together with imagined transitions generated by the learned world model. Specifically, at each update, we sample a mini-batch of real transitions $d$ from $\mathcal{D}$ and use $W_\phi$ to generate a set of context-conditioned imagined transitions $\hat{d}_\phi$ through rollout. We then form the augmented transition batch $d_f \doteq d \cup \hat{d}_\phi$, where the relative amount of imagined data is determined by the number of imagination rollouts and the rollout horizon $H$. The conservative meta-value objective is formulated as:

$$\mathcal{L}_{\text{critic}}(\omega) \doteq \beta\left(\mathbb{E}_{s,a\sim\rho}[Q_\omega(s, z, a)] - \mathbb{E}_{s,a\sim\mathcal{D}}[Q_\omega(s, z, a)]\right)$$
$$+ \frac{1}{2}\mathbb{E}_{s,a\sim d_f}\left[\left(Q_\omega(s, z, a) - \widehat{\mathcal{B}}^\pi Q_\omega(s, z, a)\right)^2\right], \qquad (9)$$

where $\rho(s, a) = d^\pi_{\widehat{\mathcal{M}}}(s)\pi_\theta(a|s, z)$ represents the state-action marginal distribution induced by the current meta-policy $\pi_\theta$ when it is executed in the learned world model $\widehat{\mathcal{M}}$. The first term actively lowers the $Q$-values of states-actions pairs that are evaluated as having high value by the current policy but are not present in the real offline data. To ensure the critic remains grounded in the real experience distribution $\mathcal{D}$, we need to increase the $Q$-values of the state-action pairs in $\mathcal{D}$, thereby preventing the meta-policy from using the imagined data generated by the world model beyond the support region and generating overly high value estimates. The second term ensures the Bellman consistency on the augmented data $d_f$, which consists of

both real transitions and imagined transitions. The operator $\widehat{\mathcal{B}}^\pi$ is the context-dependent Bellman operator, defined as $\widehat{\mathcal{B}}^\pi Q_\omega(s, z, a) \doteq r + \gamma \mathbb{E}_{a' \sim \pi_\theta(\cdot|s',z)}[Q_{\bar{\omega}}(s', z, a')]$, where $\bar{\omega}$ denotes the target network parameters.

**Meta-Policy Improvement.** Finally, under the constraints of the conservative critic, we optimize the meta-policy parameters $\theta$ by minimizing the following objective on the augmented data $d_f$ :

$$\mathcal{L}_{\text{actor}}(\theta) \doteq -\mathbb{E}_{\substack{s \sim d_f \\ a \sim \pi_\theta(\cdot|s,z)}} [Q_\omega(s, z, a)] - \alpha\mathcal{H}(\pi_\theta(\cdot|s, z)). \tag{10}$$

The conservative critic acts as an implicit regularizer for policy improvement. Since in the regions lacking data support, $Q_\omega$ is tuned to be pessimistic, maximizing it is equivalent to implicitly setting a safety constraint: the policy naturally avoids taking actions that lead to unreliable imagined states, because these states are assigned lower values during the value evaluation stage. Therefore, the agent can safely leverage the generalization ability of the world model to discover the optimal trajectory for the given task $M_i$, even in sparse-reward environments where the offline dataset may only provide suboptimal or fragmented evidence. This enables the agent to mitigate the inherent pattern dilemma in the offline setting: it no longer merely imitates the behavior policy in $\mathcal{D}$, but learns a general adaptive policy that can effectively interpolate and extrapolate on the task manifold, thereby achieving higher stability and performance in unseen tasks.

### 3.3. Theoretical Analysis of Behavior-invariant Representation in World Model

By extracting latent task dynamics from the world model, we theoretically show that such dynamics substantially reduce the influence of behavior-induced action distributions on task identification, thereby improving robustness under heterogeneous behavior policies.

**Theorem 3.1** (World model dynamics as primary causality). *Given a stochastic world model with latent states $e_t$ and temporal context embeddings $h_t = f_\phi(e_{1:t}, a_{1:t})$, $X_b = \{e_t, a_t\}$ denote behavior-related variables and $X_t = \{e_{t+1}\}$ denote task-related variables. Let $Z$ be a global task representation aggregated from $\{h_t\}$. Then optimizing the latent dynamics of the world model $\mathcal{L}_{\text{dyn}} = \mathbb{E}[\text{KL}(q_\phi(e_{t+1} \mid o_{t+1}) \| p_\phi(\hat{e}_{t+1} \mid h_t))]$ can be viewed as maximizing the primary causality $I(Z; X_t \mid X_b)$.*

This theorem shows that optimizing the latent dynamics of a world model effectively captures task-relevant primary causality in offline meta-RL settings.

**Proposition 3.2.** *For each task $M$ and behavior policy $\pi$, the task-token posterior is calibrated: $\mathbb{E}_{\tau \sim p(\tau|M,\pi)}\Big[\text{KL}\big(q_\phi(z \mid \tau)\|p(z \mid M)\big)\Big] \leq \varepsilon$, which is enforced by minimizing $\mathcal{L}_{\text{dyn}}$.*

**Theorem 3.3** (Dynamics-induced behavior invariance)**.** *Consider the task latent $Z$ inferred by the world model task token. Under standard posterior–prior calibration of the dynamics KL, if $\mathcal{L}_{\text{dyn}} \leq \varepsilon$, then $I(Z; X_b \mid M) \leq \varepsilon$.*

**Theorem 3.4** (Behavior-policy robustness of task representation)**.** *Under the same condition, for any two behavior policies $\pi, \pi'$ on task $M$, the induced representation distributions satisfy $\text{TV}\big(p(Z \mid M, \pi), p(Z \mid M, \pi')\big) \leq \sqrt{2\varepsilon}$.*

All proofs are provided in Appendix A.

## 4. Experiments

In this section, we evaluate MetaSTAR on multiple dense-reward and challenging sparse-reward tasks, aiming to address the following three core questions: 1. Does MetaSTAR outperform existing state-of-the-art (SOTA) methods in standard offline meta-reinforcement learning benchmarks, especially in sparse reward settings? 2. Can MetaSTAR maintain effective adaptation capabilities when the distribution of test tasks significantly differs from the training data distribution? 3. Can the proposed behavior-invariant representation learning framework with transformer-based world models effectively decouple task dynamics from behavior policies and generate a structured latent space?

### 4.1. Experimental Settings

**Environments and Baselines.** The experimental environments include Point-Robot-Sparse and continuous control tasks based on the MuJoCo (Todorov et al., 2012) physics engine, covering Ant-Dir, Cheetah-Vel, Hopper-Param, Walker-Param, and the sparse-reward variants with the exception of Ant-Dir (see Appendix F for details). We compare MetaSTAR against four representative context-based offline meta-reinforcement learning baseline methods: **FOCAL** (Li et al., 2020), **CORRO** (Yuan & Lu, 2022), **CSRO** (Gao et al., 2023) and **UNICORN** (Li et al., 2024). Details of these baselines are provided in Appendix E.

**Data Collection.** For each environment, we randomly sample 40 different tasks, partitioning them into a set of 30 tasks for offline meta-training, and the remaining 10 tasks for testing. For each task $M_i$, the offline dataset is constructed from the replay buffer accumulated throughout the training of an independently trained SAC agent (Haarnoja et al., 2018). Thus, each task-specific buffer contains mixed-proficiency transitions induced by a non-stationary sequence of behavior policies, denoted as $\{\pi_{\beta_i}^n\}_{n=1}^{N_{\text{update}}}$, from early exploration to more proficient policies. To evaluate the robustness of MetaSTAR in **out-of-distribution (OOD)** scenarios, we divided the data of four environments (Ant-Dir, Point-Robot-Sparse, Cheetah-Vel, and Cheetah-Vel-Sparse) into in-distribution (ID) and OOD tasks based on the task goals (see Table 4 in Appendix F).

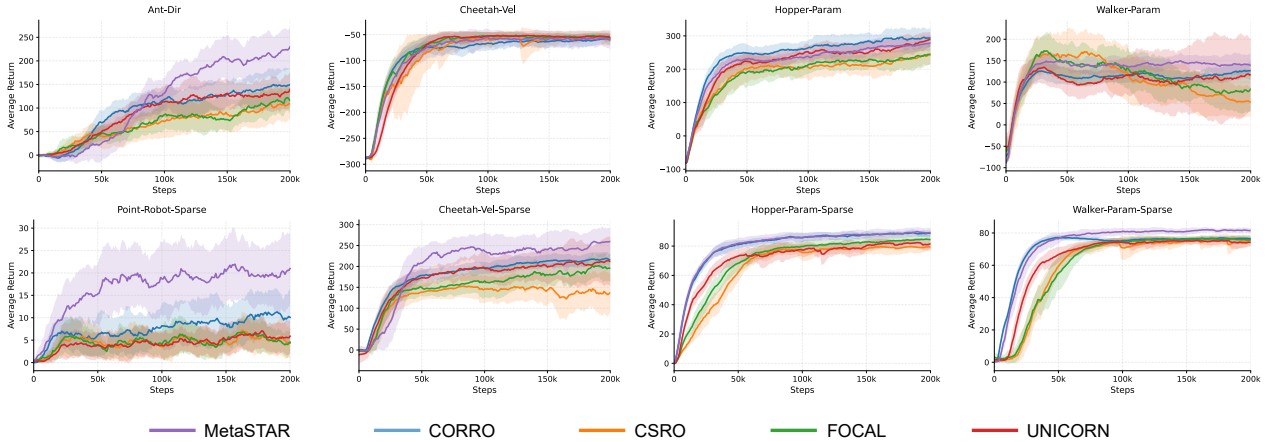

*Figure 3.* Average online meta-testing performance on 4 dense-reward environments and 4 sparse-reward environments.

**Evaluation Protocols.** In the meta-testing phase, we employ two evaluation protocols: offline test and online test. Offline test is an idealized but impractical evaluation method. It directly uses pre-collected offline data as the context, thus ignoring the context shift problem, and usually achieves higher performance. In contrast, online test involves the agent collecting context data through online interactions with the test environment using its current exploration policy. As this represents a more practical evaluation protocol, we place greater emphasis on MetaSTAR's performance during **online testing**. The results are averaged over 3 random seeds. In the tables, the best result is **bolded**, and the second-best result is underlined.

### 4.2. Main Results on Meta-Testing

Table 1 and Figure 3 present the average returns of MetaS-TAR compared to baseline methods in multiple environments. The experimental results show that MetaSTAR achieves superior performance in the vast majority of tasks. In the two dense-reward tasks of Ant-Dir and Walker-Param, the offline and online testing returns of MetaSTAR are significantly higher than those of the baselines. Notably, in Ant-Dir, MetaSTAR achieves an online return of $230 \pm 42$, while the suboptimal baseline CORRO was only $149 \pm 39$. The advantage of MetaSTAR is even more pronounced in **sparse-reward environments**, where baseline methods frequently succumb to pattern collapse. Particularly in Cheetah-Vel-Sparse and Point-Robot-Sparse, MetaSTAR achieves high returns of $296 \pm 19$ and $21 \pm 8$ respectively, far exceeding the closest baseline CORRO ($216 \pm 15$ and $10 \pm 6$). This validates that by maximizing the primary causality, MetaS-TAR can extract features intrinsic to the task dynamics from the sparse rewards, rather than overfitting to the behavior-induced correlations in the offline data. This indicates that MetaSTAR can effectively mitigates the context distribution shift and achieves more robust online adaptation.

### 4.3. Out-of-Distribution Adaptation

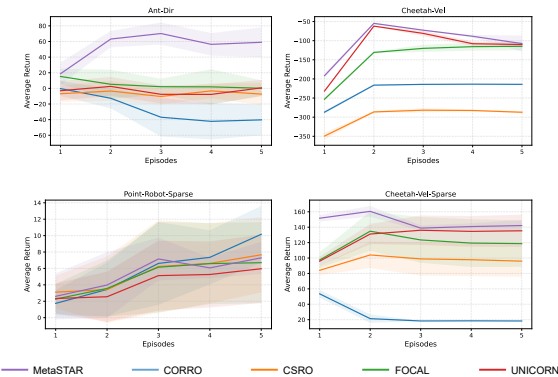

*Figure 4.* Average online adaptation performance of the first five episodes on **out-of-distribution** tasks.

To address the second question, we evaluate MetaSTAR under out-of-distribution online adaptation, where policy distribution shift becomes more challenging. Table 2 details the final episode performance of each baseline in both ID and OOD scenarios for online adaptation. The relevant adaptation curves are presented in Figure 4 (OOD) and Figure 8 (ID). From Figure 4, we can observe that in the OOD setting, MetaSTAR rapidly attains high performance within the first episode of OOD exploration and maintains stability thereafter. This indicates its ability to rapidly infer task-relevant information with limited online interaction. This success stems from the synergy between our conservative value regularization and the imagination rollouts of the world model, which enables the policy to explore safely when encountering unknown states, thereby preventing policy collapse due to overestimation of value. Conversely, while baselines like CORRO perform adequately in ID settings, their performance deteriorates significantly in OOD scenarios. These results suggest that MetaSTAR maintains stronger online adaptation ability under task distribution shifts, rather than

*Table 1.* Average meta-testing returns of MetaSTAR against other baselines.

| Environment | CORRO | | CSRO | | FOCAL | | UNICORN | | MetaSTAR | |
|---|---|---|---|---|---|---|---|---|---|---|
| | Offline | Online | Offline | Online | Offline | Online | Offline | Online | Offline | Online |
| Ant-Dir | $183 \pm 41$ | $\underline{149 \pm 39}$ | $141 \pm 42$ | $109 \pm 33$ | $194 \pm 42$ | $115 \pm 27$ | $\underline{210 \pm 40}$ | $140 \pm 36$ | $\mathbf{236 \pm 48}$ | $\mathbf{230 \pm 42}$ |
| Cheetah-Vel | $-34 \pm 3$ | $\overline{-60 \pm 10}$ | $-38 \pm 10$ | $\underline{-57 \pm 14}$ | $-36 \pm 6$ | $-57 \pm 7$ | $\underline{-32 \pm 3}$ | $\mathbf{-56 \pm 10}$ | $-33 \pm 4$ | $-60 \pm 19$ |
| Hopper-Param | $\mathbf{300 \pm 29}$ | $\mathbf{294 \pm 28}$ | $236 \pm 22$ | $\underline{242 \pm 31}$ | $253 \pm 35$ | $245 \pm 30$ | $\underline{296 \pm 24}$ | $289 \pm 22$ | $276 \pm 24$ | $279 \pm 25$ |
| Walker-Param | $\underline{126 \pm 25}$ | $\underline{127 \pm 22}$ | $28 \pm 33$ | $54 \pm 34$ | $116 \pm 31$ | $84 \pm 50$ | $\overline{103 \pm 30}$ | $\overline{117 \pm 86}$ | $\mathbf{140 \pm 24}$ | $\mathbf{139 \pm 27}$ |
| Point-Robot-Sparse | $38 \pm 7$ | $\underline{10 \pm 6}$ | $13 \pm 7$ | $4 \pm 3$ | $28 \pm 5$ | $5 \pm 3$ | $\underline{40 \pm 5}$ | $6 \pm 3$ | $\mathbf{42 \pm 5}$ | $\mathbf{21 \pm 8}$ |
| Cheetah-Vel-Sparse | $\underline{260 \pm 13}$ | $\underline{216 \pm 15}$ | $169 \pm 59$ | $137 \pm 46$ | $242 \pm 18$ | $196 \pm 31$ | $\overline{238 \pm 14}$ | $212 \pm 60$ | $\mathbf{296 \pm 19}$ | $\mathbf{260 \pm 34}$ |
| Hopper-Param-Sparse | $\underline{89 \pm 2}$ | $\underline{89 \pm 2}$ | $83 \pm 4$ | $79 \pm 3$ | $88 \pm 3$ | $84 \pm 4$ | $85 \pm 3$ | $81 \pm 4$ | $\mathbf{90 \pm 4}$ | $\mathbf{89 \pm 3}$ |
| Walker-Param-Sparse | $\underline{78 \pm 1}$ | $\underline{76 \pm 1}$ | $75 \pm 4$ | $75 \pm 3$ | $76 \pm 3$ | $76 \pm 2$ | $75 \pm 5$ | $74 \pm 2$ | $\mathbf{81 \pm 2}$ | $\mathbf{82 \pm 2}$ |

*Table 2.* Average online adaptation performance for the final episode.

| Environment | CORRO | | CSRO | | FOCAL | | UNICORN | | MetaSTAR | |
|---|---|---|---|---|---|---|---|---|---|---|
| | ID | OOD | ID | OOD | ID | OOD | ID | OOD | ID | OOD |
| Ant-Dir | $136 \pm 34$ | $-40 \pm 20$ | $143 \pm 30$ | $-7 \pm 14$ | $75 \pm 16$ | $0 \pm 10$ | $\underline{200 \pm 24}$ | $\underline{1 \pm 10}$ | $\mathbf{269 \pm 24}$ | $\mathbf{59 \pm 19}$ |
| Cheetah-Vel | $-76 \pm 13$ | $-214 \pm 1$ | $-223 \pm 17$ | $-287 \pm 3$ | $-164 \pm 45$ | $-115 \pm 10$ | $\underline{-68 \pm 1}$ | $\underline{-110 \pm 7}$ | $\mathbf{-53 \pm 1}$ | $\mathbf{-107 \pm 21}$ |
| Hopper-Param | $302 \pm 19$ | / | $257 \pm 10$ | / | $259 \pm 19$ | / | $259 \pm 16$ | / | $\underline{270 \pm 12}$ | / |
| Walker-Param | $\underline{119 \pm 13}$ | / | $63 \pm 17$ | / | $53 \pm 4$ | / | $47 \pm 6$ | / | $\mathbf{158 \pm 6}$ | / |
| Point-Robot-Sparse | $\underline{17 \pm 5}$ | $\mathbf{10 \pm 3}$ | $2 \pm 2$ | $8 \pm 5$ | $5 \pm 2$ | $7 \pm 5$ | $4 \pm 3$ | $6 \pm 4$ | $\mathbf{36 \pm 5}$ | $7 \pm 2$ |
| Cheetah-Vel-Sparse | $214 \pm 26$ | $18 \pm 1$ | $183 \pm 8$ | $\underline{96 \pm 20}$ | $190 \pm 15$ | $119 \pm 30$ | $\underline{240 \pm 19}$ | $135 \pm 21$ | $\mathbf{269 \pm 15}$ | $\mathbf{142 \pm 7}$ |
| Hopper-Param-Sparse | $75 \pm 1$ | / | $\underline{77 \pm 1}$ | / | $76 \pm 1$ | / | $74 \pm 1$ | / | $\mathbf{81 \pm 1}$ | / |
| Walker-Param-Sparse | $\underline{88 \pm 1}$ | / | $\overline{70 \pm 1}$ | / | $87 \pm 2$ | / | $80 \pm 1$ | / | $\mathbf{89 \pm 1}$ | / |

overfitting to the offline training distribution.

## 4.4. Visualization of Task Representations

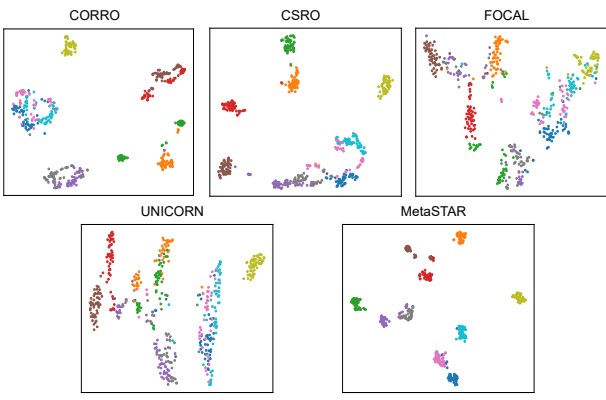

*Figure 5.* T-SNE visualization of the learned task representation space in Hopper-Param.

To qualitatively examine the property of **behavior invariance**, we used t-SNE (Maaten & Hinton, 2008) to visualize the learned task representations. Figure 5 shows the latent space structures of different methods in Hopper-Param. As clearly shown in this figure, the task representations generated by MetaSTAR exhibit highly structured clustering characteristics. Samples from the same task are closely clustered, while the boundaries between different tasks are distinct. In contrast, the representations of other baselines are more disordered and even overlap between different tasks. This suggests that their task representations are susceptible to the influence of behavior. This support our theoretical analysis that the Transformer-based world model successfully extracts the behavior-invariant task representa-

tations. Additional t-SNE visualizations are provided in Appendix D.3, including dense-reward environments in Figure 9 and sparse-reward environments in Figure 10.

## 4.5. Euclidean Distance of Task Representations

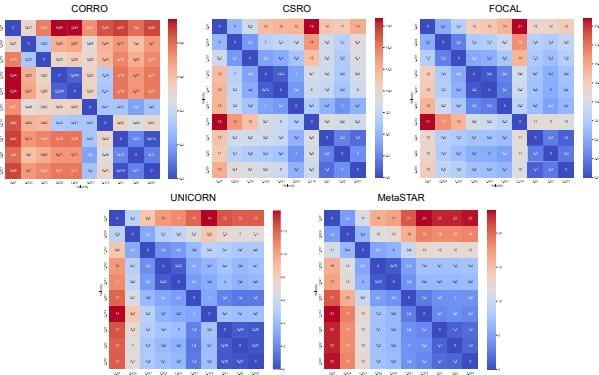

*Figure 6.* Euclidean distance of task representations on Cheetah-Vel-Sparse.

Beyond t-SNE visualization, we further analyze whether the learned representation space preserves the semantic relationships among unseen tasks. Specifically, we compute the pairwise Euclidean distances between task embeddings and compare the resulting distance patterns with the underlying task parameters. Figure 6 shows the distance matrices on Cheetah-Vel-Sparse, where the task goal is the target velocity sampled from [0.0, 3.0]. MetaSTAR exhibits a clear locally smooth structure: tasks with similar target velocities are mapped closer in the latent space, while tasks with larger velocity gaps tend to have larger representation distances. This trend remains visible despite the sparse-reward

setting, indicating that MetaSTAR can extract task-relevant structure from limited feedback. In contrast, the baseline methods show weaker or less consistent distance patterns, suggesting that their representations are less aligned with the underlying task semantics. These results complement the t-SNE visualizations and provide further evidence that MetaSTAR encourages task representations to capture task-relevant structure while reducing behavior-induced variations. Additional Euclidean distance analyses on Cheetah-Vel and Point-Robot-Sparse are provided in Appendix D.4.

### 4.6. Ablation Study

We conduct ablation studies to examine the contribution of the world-model objective $\mathcal{L}_{\text{WM}}$ and conservative policy optimization. Figure 13 and Figure 14 compare the full MetaSTAR with its ablated variants under online and offline testing. Removing $\mathcal{L}_{\text{WM}}$ generally degrades performance, confirming the importance of world-model-based representation learning for task inference and cross-task generalization. Removing conservative policy optimization also hurts performance in many settings, indicating its role in reducing value overestimation and limiting model exploitation during imagination-based policy learning. When both components are removed, the model becomes a simplified metric-learning variant without latent dynamics learning or conservative imagination-based policy optimization, leading to the largest degradation, especially in sparse-reward environments. These results show that the two components are complementary: $\mathcal{L}_{\text{WM}}$ improves task representation under context distribution shift, while conservative policy optimization stabilizes policy learning under policy distribution shift. We also observe task-dependent effects; for example, on Hopper-Param, removing conservative policy optimization slightly improves performance, possibly because dense rewards and informative offline trajectories reduce the need for imagined data, while model rollouts may introduce additional bias.

### 5. Related Works

**Task Representation Learning in Meta-RL.** Learning compact and expressive task representations is central to meta-RL, as it enables agents to extract task-relevant information and rapidly adapt corresponding tasks (Beck et al., 2025). Prior works explore different representation methods in meta-RL, including reconstruction-based inference (Zintgraf et al., 2020; 2021), distance metric learning (Li et al., 2020), contrastive learning (Yuan & Lu, 2022; Choshen & Tamar, 2023) and bisimulation metrics (Zhang et al., 2021; Zang et al., 2023; Zhang et al., 2025b).

**Information-theoretic Perspectives on Task Inference.** Information-theoretic and causal perspectives have been widely used to study representation learning, invariance,

and generalization (Schölkopf et al., 2021; Mu et al., 2022; Wang et al., 2022; Yu et al., 2024). In offline meta-RL, CORRO maximizes task-related information with contrastive learning over transition tuples (Yuan & Lu, 2022), while CSRO reduces context shift by suppressing policy-related information in task representations (Gao et al., 2023). UNICORN further provides a unified information-theoretic framework that interprets FOCAL, CORRO, and CSRO as different approximations of task information (Li et al., 2024). BATI performs behavior-agnostic task inference through dynamics-based likelihood (Ma et al., 2025), while recent work analyzes the impact of task representation shift during encoder-policy optimization (Zhang et al., 2025a).

**World Models.** World models learn stochastic latent dynamics and enable imagination-based planning under partial observability (Hafner et al., 2019; 2020; 2021; 2025a;b). Transformer-based world models further improve long-range dependency modeling and complex dynamics prediction (Micheli et al., 2022; Robine et al., 2023; Zhang et al., 2023; Chen et al., 2022; Burchi & Timofte, 2025). In meta-RL, MAMBA directly instantiates a Dreamer-style learning-and-imagination pipeline for task adaptation, and MuDreamer explores robustness by relaxing reconstruction-centric training (Rimon et al., 2024; Burchi & Timofte, 2024).

### 6. Conclusions

We present MetaSTAR, a novel offline meta-RL framework for robust offline-to-online adaptation under sparse-reward and out-of-distribution settings. MetaSTAR uses a stochastic Transformer-based world model to learn behavior-invariant task representations that capture task-relevant dynamics while reducing variations induced by heterogeneous behavior policies. It further combines contextual imagination with conservative policy optimization to leverage imagined transitions while mitigating value overestimation and model exploitation in unsupported regions. Our theoretical analysis and empirical results suggest that world-model-based representation learning provides an effective way to address context and policy distribution shifts in offline meta-RL. We believe MetaSTAR offers a promising direction for robust task adaptation and efficient transfer from offline datasets to online deployment in real-world environments.

**Limitation.** MetaSTAR introduces additional computational overhead compared with model-free offline meta-RL baselines, mainly due to stochastic Transformer-based world-model training and contextual imagination during policy optimization. Although this cost is incurred entirely during offline training and does not substantially affect test-time action inference, it may limit scalability to very large task collections or high-dimensional observation domains. A detailed training-time analysis is provided in Appendix D.11.

## Acknowledgements

This work was supported by Brain Science and Brain-like Intelligence Technology - National Science and Technology Major Project (2021ZD0200500), the National Natural Science Foundation of China (62472206, 3254100307), National Key R&D Program of China (2025YFC3410000), Shenzhen Science and Technology Innovation Committee (RCYX20231211090405003, JCYJ20220818100213029), Guangdong Basic and Applied Basic Research Foundation (2026B1515020099), Guangdong S&T Program (Grant No. 2026B0101110003), Shanghai Municipal Special Program for Basic Research on General AI Foundation Models (2025SHZDZX026D05), GuangDong Basic and Applied Basic Research Foundation (2025A1515011645 to ZC.L.), GuangDong Basic and Applied Basic Research Foundation (2026A1515010121 to XK.S.), Shenzhen Doctoral Startup Project (RCBS20231211090748082 to XK.S.), Shenzhen Loop Area Institute under grant FPF10120250012, and the open research fund of the Guangdong Provincial Key Laboratory of Mathematical and Neural Dynamical Systems, the Center for Computational Science and Engineering at Southern University of Science and Technology, Shenzhen Key Laboratory of Smart Healthcare Engineering.

## Impact Statement

This research aims to promote the development of offline meta-reinforcement learning. MetaSTAR learns behavior-invariant task representations from static datasets and employs a world model for planning within the latent space. This strategy effectively expands the support region of the offline datasets, and demonstrates a stronger generalization ability in sparse-reward and out-of-distribution scenarios. Therefore, our method can significantly reduce the need for high-risk and costly real-world exploration during the training stage. These advancements can lower the deployment costs and safety risks in fields such as industrial robots, healthcare, and autonomous driving, as conducting online trial-and-error is often not feasible in these domains.

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

# A. Theorems and Proofs

**Theorem A.1** (World model dynamics as primary causality). *Given a stochastic world model with latent states $e_t$ and temporal context embeddings $h_t = f_\phi(e_{1:t}, a_{1:t})$, $X_b = \{e_t, a_t\}$ denote behavior-related variables and $X_t = \{e_{t+1}\}$ denote task-related variables. Let $Z$ be a global task representation aggregated from $\{h_t\}$. Then optimizing the latent dynamics of the world model $\mathcal{L}_{\mathrm{dyn}} = \mathbb{E}[\mathrm{KL}(q_\phi(e_{t+1} \mid o_{t+1}) \,\|\, p_\phi(\hat{e}_{t+1} \mid h_t))]$ can be viewed as maximizing the primary causality $I(Z; X_t \mid X_b)$.*

*Proof.* In world models, the observation $o_t = [s_t, r_{t-1}]$ is encoded into the stochastic latent states $e_t$. Therefore, the behavior-related components is $X_b = \{e_t, a_t\}$ and the task-related components is $X_t = \{e_{t+1}\}$. As $Z$ is global task representation distilled from $\{h_1, \ldots, h_T\}$ via self-attention, maximizing information about the environment's physics $I(Z; X_t | X_b)$ is equivalent to maximizing the mutual information between the hidden state and the future latent state:

$$I(Z; X_t | X_b) = I(h_t; X_t | X_b). \tag{11}$$

Using the chain rule for mutual information:

$$I(h_t; X_t | X_b) = I(X_t; h_t, X_b) - I(X_t; X_b). \tag{12}$$

In the offline setting, the dataset is fixed. $I(X_t; X_b)$ depends only on the environment, rendering it a constant relative to the optimization of the dynamics model. Let $\equiv$ denote equality up to a constant, then

$$I(h_t; X_t | X_b) \equiv I(X_t; h_t, X_b). \tag{13}$$

Using the chain rule:

$$I(X_t; h_t, X_b) = I(X_t; h_t) + I(X_t; X_b | h_t). \tag{14}$$

By architecture, the dynamics predictor $p_\phi$ only receives $h_t$ as input. The raw history $X_b$ provides no additional information about the future $e_{t+1}$ once the context embedding $h_t$ is computed. Thus, $X_b \to h_t \to X_t$ forms a Markov Chain, implying $I(X_t; X_b | h_t) = 0$. Thus, we have:

$$I(h_t; X_t | X_b) \equiv I(X_t; h_t). \tag{15}$$

Expanding the mutual information:

$$\begin{aligned} I(X_t; h_t) &= H(X_t) - H(X_t | h_t) \\ &= H(e_{t+1}) - H(e_{t+1} | h_t). \end{aligned} \tag{16}$$

Since the marginal entropy of the latent states $H(e_{t+1})$ is determined by the encoder and the fixed dataset, maximizing the mutual information is equivalent to minimizing the conditional entropy $H(e_{t+1} | h_t)$:

$$\max I(h_t; X_t | X_b) \iff \min H(e_{t+1} | h_t). \tag{17}$$

The true conditional entropy $H(e_{t+1} | h_t)$ is intractable. We use the world model's prior distribution $p_\phi(\hat{e}_{t+1} | h_t)$ as a variational approximation. By Gibbs' Inequality, the cross-entropy serves as an upper bound:

$$H(e_{t+1} | h_t) \leq \mathbb{E}_{e \sim q_\phi, h \sim f_\phi}[-\log p_\phi(e_{t+1} | h_t)]. \tag{18}$$

In the world model objective, this is minimized via the KL Divergence (Dynamics Loss):

$$\mathcal{L}_{\mathrm{dyn}} = \mathbb{E}[\mathrm{KL}(q_\phi(e_{t+1} | o_{t+1}) \,\|\, p_\phi(\hat{e}_{t+1} | h_t))] = \mathbb{E}_q[-\log p_\phi(e_{t+1} | h_t)] - H(q_\phi). \tag{19}$$

Since the posterior entropy $H(q_\phi)$ is independent of the dynamics predictor, minimizing $\mathcal{L}_{\mathrm{dyn}}$ directly minimizes the variational upper bound of the conditional entropy, thereby maximizing the primary causality $I(Z; X_t | X_b)$. That is:

$$\min \mathcal{L}_{\mathrm{WM}} \Rightarrow \min \mathcal{L}_{\mathrm{dyn}} \Rightarrow \max I(Z; X_t \mid X_b). \tag{20}$$

$\square$

**Proposition A.2.** *Conditioned on a fixed task $M$, the behavior policy $\pi$ influences the sampled segment $\tau$ only through $X_b$ (state-action visitation), and $Z$ is inferred from $\tau$. Hence, the following Markov chain holds:*

$$\pi \;\to\; X_b \;\to\; \tau \;\to\; Z \qquad (\textit{conditioned on } M). \tag{21}$$

**Proposition A.3.** *There exists a task-conditional reference distribution $p(z \mid M)$ such that for any behavior policy $\pi$,*

$$\mathbb{E}_{\tau \sim p(\tau \mid M, \pi)}\Big[\mathrm{KL}\big(q_\phi(z \mid \tau) \,\|\, p(z \mid M)\big)\Big] \;\leq\; \varepsilon. \tag{22}$$

*Moreover, minimizing $\mathcal{L}_{\mathrm{dyn}} = \mathbb{E}[\mathrm{KL}(q_\phi(e_{t+1} \mid o_{t+1}) \,\|\, p_\phi(\hat{e}_{t+1} \mid h_t))]$ ensures (22) holds with $\varepsilon$ upper bounded by $\mathcal{L}_{\mathrm{dyn}}$ (up to a task-dependent constant absorbed into $\varepsilon$).*

**Remark.** Proposition A.3 is the standard "posterior calibration" condition in latent-variable world models: the inferred task latent from a segment is close to a task-identity distribution $p(z \mid M)$. In our architecture, $\mathcal{L}_{\mathrm{dyn}}$ enforces posterior–prior consistency ($q_\phi(e_{t+1} \mid o_{t+1})$ vs. $p_\phi(\hat{e}_{t+1} \mid h_t)$); under a well-specified model and sufficient optimization, the induced posteriors cluster around a task-specific distribution, yielding (22).

**Theorem A.4** (Dynamics-induced behavior invariance). *Under Proposition A.2–A.3, if $\mathcal{L}_{\mathrm{dyn}} \leq \varepsilon$, then the lesser causality of the task representation is bounded as*

$$I(Z; X_b \mid M) \;\leq\; \varepsilon. \tag{23}$$

*Equivalently, the task representation $Z$ is $\varepsilon$-invariant to behavior-induced context $X_b$ under a fixed task.*

*Proof.* Since $X_b$ is a function of the trajectory segment $\tau$, by data processing,

$$I(Z; X_b \mid M) \;\leq\; I(Z; \tau \mid M). \tag{24}$$

We next upper bound $I(Z; \tau \mid M)$ using the calibration condition (22). Using the identity

$$I(Z; \tau \mid M) = \mathbb{E}_{\tau \sim p(\tau \mid M, \pi)}\Big[\mathrm{KL}\big(q_\phi(z \mid \tau) \,\|\, p(z \mid M)\big)\Big] - \mathrm{KL}\big(q_\phi(z \mid M) \,\|\, p(z \mid M)\big), \tag{25}$$

where $q_\phi(z \mid M) = \mathbb{E}_{\tau \sim p(\tau \mid M, \pi)}[q_\phi(z \mid \tau)]$ is the aggregated posterior under task $M$. Since KL is nonnegative, we obtain

$$I(Z; \tau \mid M) \leq \mathbb{E}_{\tau \sim p(\tau \mid M, \pi)}\Big[\mathrm{KL}\big(q_\phi(z \mid \tau) \,\|\, p(z \mid M)\big)\Big] \leq \varepsilon. \tag{26}$$

Combining yields $I(Z; X_b \mid M) \leq \varepsilon$. Finally, by Proposition A.3, minimizing $\mathcal{L}_{\mathrm{dyn}}$ ensures the above holds with $\varepsilon$ upper bounded by $\mathcal{L}_{\mathrm{dyn}}$. $\qquad\square$

**Theorem A.5** (Behavior-policy robustness of task representation). *Fix a task $M$ and any two behavior policies $\pi$ and $\pi'$. Let $P_M^\pi \triangleq p_\phi(z \mid M, \pi)$ and $P_M^{\pi'} \triangleq p_\phi(z \mid M, \pi')$ denote the induced distributions of $Z$. Under Proposition A.2, if $I(Z; X_b \mid M) \leq \varepsilon$, then*

$$\mathrm{TV}\big(P_M^\pi, \; P_M^{\pi'}\big) \;\leq\; \sqrt{2\varepsilon}. \tag{27}$$

*In particular, by Theorem A.4, if $\mathcal{L}_{\mathrm{dyn}} \leq \varepsilon$, then $\mathrm{TV}(P_M^\pi, P_M^{\pi'}) \leq \sqrt{2\varepsilon}$.*

*Proof.* By Proposition A.2 and data processing,

$$I(Z; \pi \mid M) \;\leq\; I(Z; X_b \mid M) \;\leq\; \varepsilon.$$

Now define a binary random variable $\Pi \in \{\pi, \pi'\}$ with $\mathbb{P}(\Pi = \pi) = \mathbb{P}(\Pi = \pi') = \frac{1}{2}$. Then $I(Z; \Pi \mid M)$ equals the Jensen–Shannon divergence between $P_M^\pi$ and $P_M^{\pi'}$ (in nats). Let $\bar{P} \triangleq \frac{1}{2}(P_M^\pi + P_M^{\pi'})$. By Pinsker's inequality,

$$\mathrm{TV}(P_M^\pi, \bar{P}) \leq \sqrt{\mathrm{KL}(P_M^\pi \| \bar{P})/2}, \qquad \mathrm{TV}(P_M^{\pi'}, \bar{P}) \leq \sqrt{\mathrm{KL}(P_M^{\pi'} \| \bar{P})/2}. \tag{28}$$

Using triangle inequality and Cauchy–Schwarz,

$$\mathrm{TV}(P_M^\pi, P_M^{\pi'}) \leq \mathrm{TV}(P_M^\pi, \bar{P}) + \mathrm{TV}(\bar{P}, P_M^{\pi'}) \leq \sqrt{2\Big(\tfrac{1}{2}\mathrm{KL}(P_M^\pi \| \bar{P}) + \tfrac{1}{2}\mathrm{KL}(P_M^{\pi'} \| \bar{P})\Big)} = \sqrt{2 I(Z; \Pi \mid M)}. \tag{29}$$

Finally $I(Z; \Pi \mid M) \leq I(Z; \pi \mid M) \leq \varepsilon$ yields $\mathrm{TV}(P_M^\pi, P_M^{\pi'}) \leq \sqrt{2\varepsilon}$. $\qquad\square$

## B. Code of MetaSTAR

Our code is available at https://github.com/QianFY/MetaSTAR

## C. MetaSTAR Pseudo-code

---
**Algorithm 1** Meta-training
---

**Require:** Offline Datasets $\mathcal{D} = \{\mathcal{D}_i\}_{i=1}^{\text{train}}$ of a set of training tasks $\boldsymbol{M} = \{M_i\}_{i=1}^{N_{\text{train}}}$, initialize learned policy $\pi_\theta$, Q-function $Q_\omega$, world model $W_\phi$

**Parameter:** $\theta, \omega, \phi$

**while** not done **do**

   Sample task batch $\{M_j\}_{j=1}^{N_{\text{batch}}}$ from $\boldsymbol{M}$

   **for** step in each iter **do**

      // World Model Learning

      Sample context $c = \{(s_t, a_t, r_t, s_t')\}_{t=1}^T$ from $\mathcal{D}$

      Get the observation $o_t = [s_t, r_{t-1}]$

      Sample latent observation by $e_t \sim q_\phi(e_t \mid o_t)$

      Compute hidden states $h_{1:T}, h_z = f_\phi(\tau_{1:T}^{e,a} = \{(e_t, a_t)\}_{t=1}^T)$

      Predict next latent observation $\hat{e}_{t+1} \sim p_\phi(\hat{e}_{t+1} \mid h_t)$

      Predict continuation $\hat{c}_t \sim p_\phi(\hat{c}_t \mid h_t)$

      Project $h_z$ to task embedding space $z \sim q_\phi(z \mid h_z)$

      Calculate $\mathcal{L}_{\text{pred}}$, $\mathcal{L}_{\text{dyn}}$ and $\mathcal{L}_{\text{rep}}$ according to Eq. (6)

      Calculate $\mathcal{L}_{\text{WM}}$ according to Eq. (5)

      Calculate $\mathcal{L}_{\text{FOCAL}}$ according to Eq. (7)

      Update $\phi$ via $\mathcal{L}_{\text{WM}}$ and $\mathcal{L}_{\text{FOCAL}}$ according to Eq. (8)

      // Agent Learning

      Sample imagination context $\{(s_t, a_t, r_t, s_t')\}_{t=1}^L$ from $\mathcal{D}$

      Imagine trajectories $\hat{d}_\phi = \{(\hat{s}_t, \hat{a}_t, \hat{r}_t, \hat{s}_t')\} \sim W_\phi$

      Sample offline data $d \sim \mathcal{D}$

      $d_f = d \cup \hat{d}_\phi$

      Conservatively evaluate $\pi_\theta$ to obtain $\widehat{\mathcal{B}}^\pi Q_\omega(s, z, a)$ using $d_f$

      Update $\omega$ to minimize $\mathcal{L}_{critic}$ according to Eq. (9)

      Update $\theta$ to minimize $\mathcal{L}_{actor}$ according to Eq. (10)

   **end for**

**end while**

**Components of world model** $W_\phi$

| | |
|---|---|
| Observation encoder: | $q_\phi(e_t \mid o_t)$ |
| Observation decoder: | $p_\phi(\hat{o}_t \mid e_t)$ |
| Sequence model: | $f_\phi(\tau_{1:t}^{e,a})$ |
| Latent dynamics predictor: | $p_\phi(\hat{e}_{t+1} \mid h_t)$ |
| Continuation predictor: | $p_\phi(\hat{c}_t \mid h_t)$ |
| Task embedding projector: | $q_\phi(z \mid h_z)$ |

---
**Algorithm 2** Meta-testing
---

**Require:** A set of testing tasks $\boldsymbol{M} = \{M_i\}_{i=1}^{N_{\text{test}}}$, learned policy $\pi_\theta$, world model $W_\phi$, initial exploration length $L_c$

**for** each task $M_i$ **do**

   Initialize context buffer $c = \text{Queue}(\text{maxlen} = L_c)$

   **for** $t = 0, ..., T - 1$ **do**

      **if** $t < L_c$ **then**

         Agent samples a random action $a_t$ to roll out $(s_t, a_t, r_t, s_t')$

      **else**

         Compute task embedding $z \sim q_\phi(z \mid h_z)$, where $h_z = W_\phi(c)$

         Agent uses $a_t \sim \pi_\theta(\cdot|s, z)$ to roll out $(s_t, a_t, r_t, s_t')$

      **end if**

      Push transition $(s_t, a_t, r_t, s_t')$ into $c$

   **end for**

   Compute episode return for evaluation

**end for**

---

# D. Additional Experiments

## D.1. Offline Meta-Testing Performance

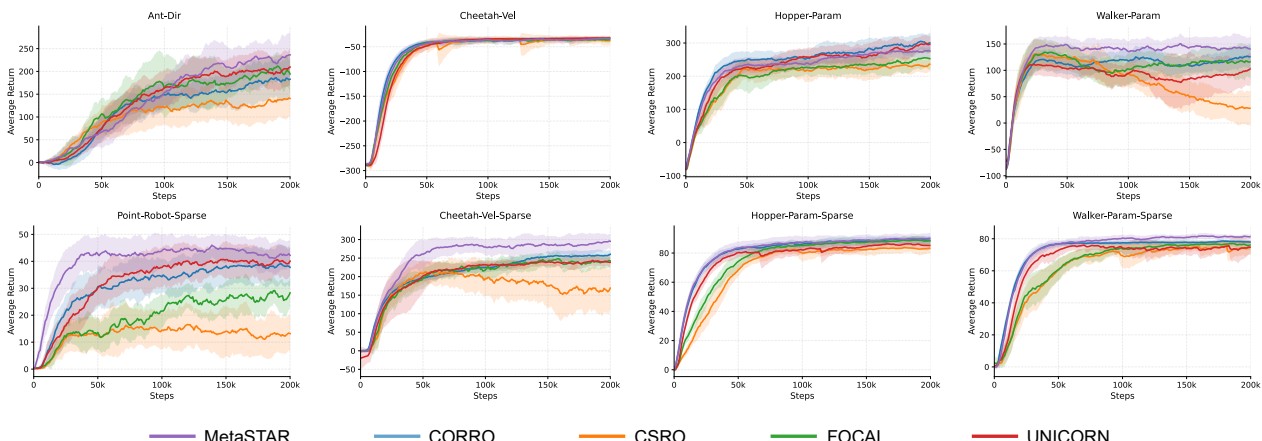

*Figure 7.* Average offline meta-testing performance on 4 dense-reward environments and 4 sparse-reward environments.

Figure 7 reports the average offline meta-testing performance on four dense-reward environments and four sparse-reward environments. This protocol allows us to quantify the performance penalty caused by the mismatch between the behavior policies used for offline data collection and the exploration policy used during online adaptation. Under offline testing, all methods generally achieve improved performance compared with online testing, since they can infer tasks from pre-collected contexts without facing online context shift. Nevertheless, MetaSTAR remains competitive or superior in most environments under this idealized setting. This indicates that its performance gain does not solely come from handling online context collection, but also from stronger task representation learning and more reliable policy optimization. Comparing the offline and online testing curves further reveals the sensitivity of different methods to the source of context. Several baselines benefit substantially from pre-collected offline contexts, but their performance degrades when they are required to infer tasks from online contexts collected by the learned policy. In contrast, MetaSTAR exhibits a smaller performance gap between offline and online testing, indicating better robustness to changes in the context-collection policy and more reliable offline-to-online adaptation.

## D.2. Online Adaptation on In-Distribution Tasks

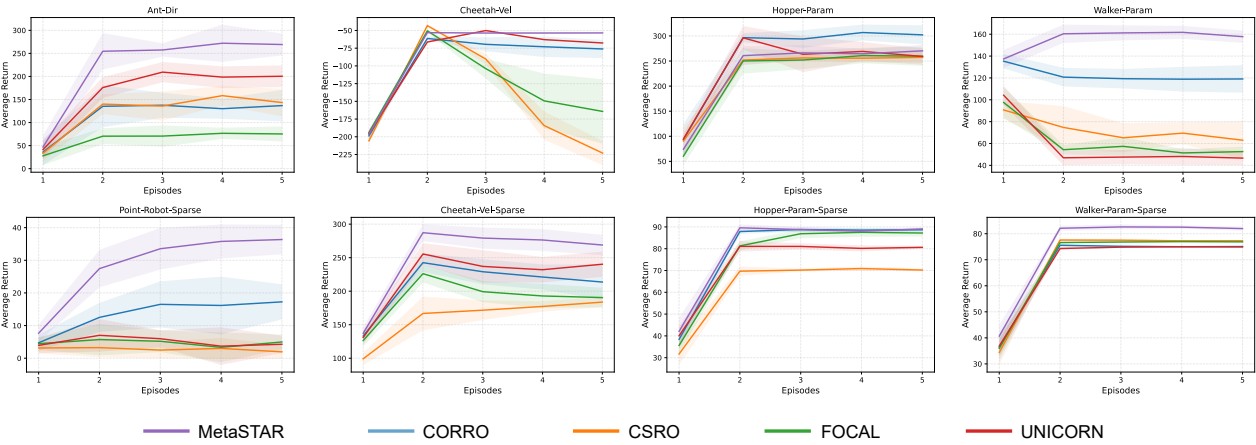

*Figure 8.* Average online adaptation performance of the first five episodes on in-distribution tasks.

Figure 8 reports the average online adaptation performance over the first five episodes on in-distribution tasks. Under the ID setting, several baselines, achieve competitive adaptation performance, indicating that their learned task representations can

handle moderate context shifts when the test tasks remain close to the training distribution. However, comparing these ID results with the OOD adaptation results in Figure 4 reveals a clear difference. Some baselines that perform well on ID tasks suffer substantial degradation under OOD tasks, suggesting that mitigating context shift alone may not be sufficient for robust offline-to-online adaptation when policy distribution shift and task distribution shift become more severe. In contrast, MetaSTAR maintains strong performance in both ID and OOD settings. This supports the effectiveness of combining behavior-invariant task representation learning with conservative policy optimization for more reliable online adaptation.

### D.3. Additional t-SNE Visualizations of Task Representations

We provide additional t-SNE visualizations of task representations under online testing in both dense-reward and sparse-reward environments. These visualizations complement the quantitative results by showing how task embeddings are organized when contexts are collected online by the agent rather than sampled from pre-collected offline data. Since online contexts may differ substantially from the offline training contexts, a well-structured embedding space indicates that the representation is less sensitive to the behavior policy and more aligned with task-relevant information.

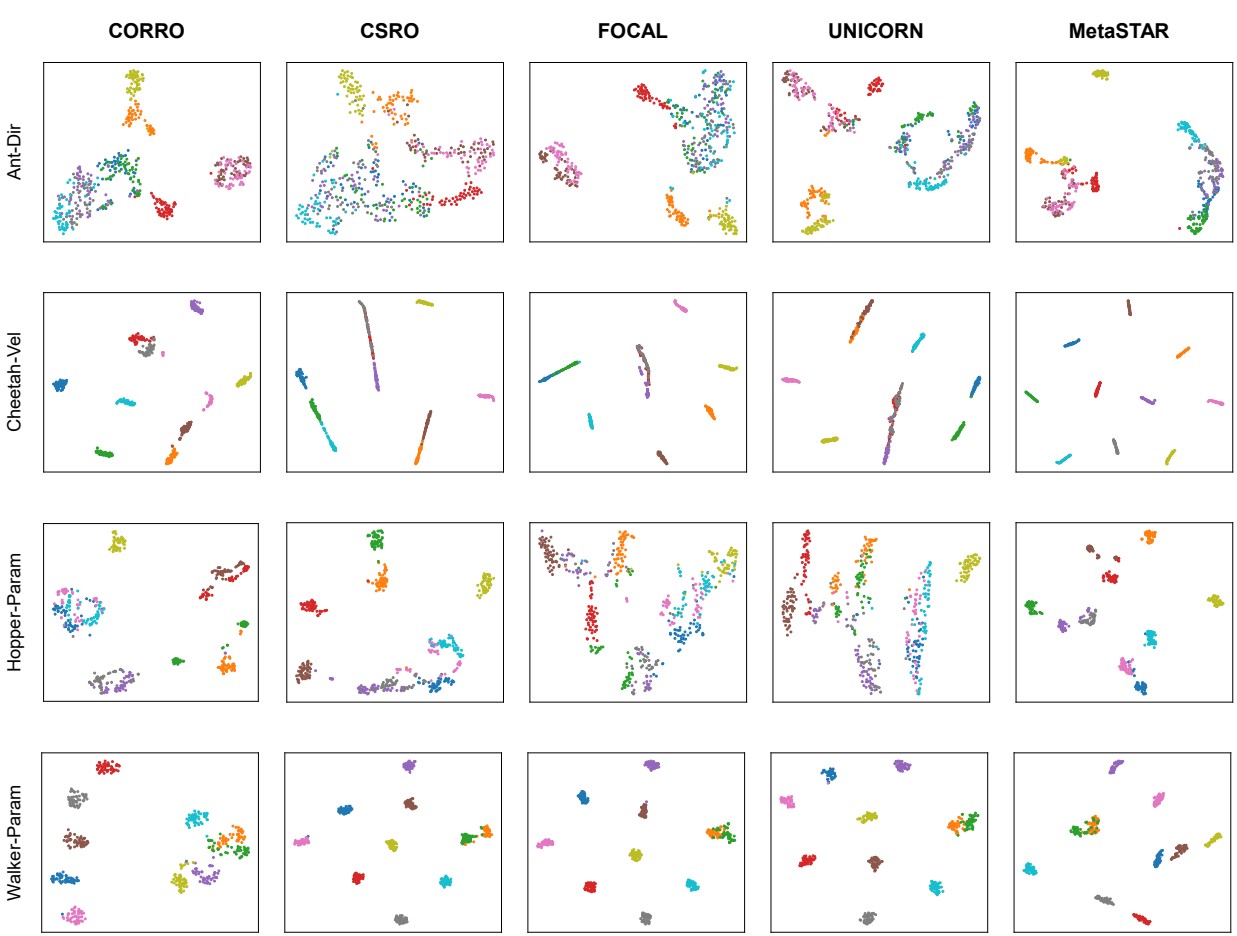

*Figure 9.* T-SNE visualization of the task representation with online testing in dense-reward environments.

Figure 9 shows the results on dense-reward environments. MetaSTAR produces structured latent clusters, where samples from the same task are generally grouped together and different tasks are better separated. This suggests that the learned representation can preserve task identity even when the context is collected through online interaction. In relatively simpler environments such as Cheetah-Vel and Walker-Param, some baselines also produce recognizable clusters, indicating that dense rewards provide sufficient task-identifying signals for these methods. However, in more challenging environments such as Ant-Dir and Hopper-Param, the baseline representations become less structured and show more overlap across tasks. This suggests that their task embeddings are more sensitive to variations in the collected online contexts and may capture behavior-induced correlations rather than task-relevant dynamics.

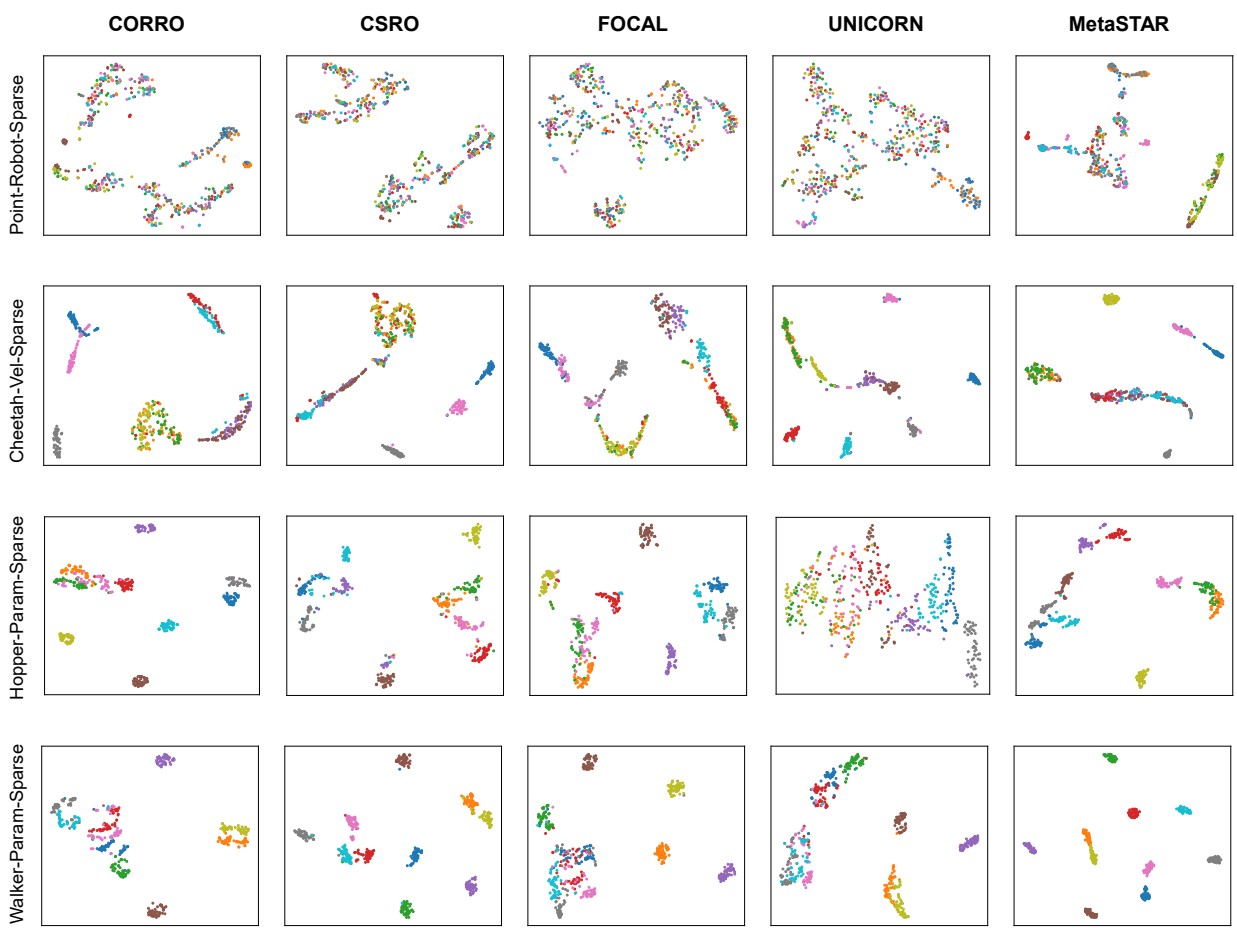

*Figure 10.* T-SNE visualization of the task representation with online testing in sparse-reward environments.

Figure 10 shows the visualization results on sparse-reward environments. Compared with dense-reward tasks, sparse-reward settings provide weaker and less frequent task-identifying signals, making task inference more challenging. Under this setting, MetaSTAR still maintains clearer cluster structures than the baselines, with embeddings from the same task grouped more consistently and different tasks better separated. In contrast, the baseline methods exhibit more overlap and less organized latent structures, suggesting that sparse feedback makes their task representations more vulnerable to context variation and behavior-induced bias.

This result is consistent with the design of MetaSTAR. The FOCAL metric objective encourages embeddings from the same task to be close and those from different tasks to be separated, while the world-model dynamics objective encourages the representation to preserve transition-related and reward-related predictive information. These two objectives provide complementary supervision: the former improves task-level discriminability, and the latter encourages dynamics-centric representations that are less dependent on behavior policies. Together, they help MetaSTAR learn more structured task representations under online testing, especially when sparse rewards make direct task identification difficult.

### D.4. Additional Euclidean Distance Analysis of Task Representations

To further examine whether the learned representation space preserves meaningful task relationships on unseen tasks, we provide additional Euclidean distance matrices of task representations on Cheetah-Vel and Point-Robot-Sparse. Unlike t-SNE visualizations, which provide a qualitative two-dimensional projection, the distance matrices directly measure pairwise distances between task embeddings in the learned representation space. Therefore, they offer a complementary view of whether the learned embeddings reflect the semantic relationships among tasks.

Figure 11 shows the results on Cheetah-Vel, where the task goal is the target velocity sampled from [0.0, 3.0]. Since

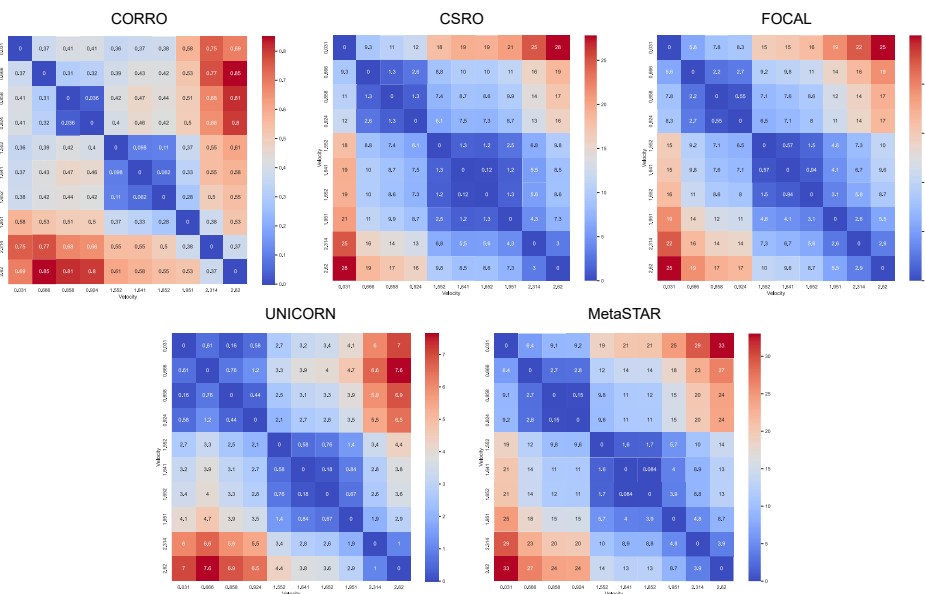

*Figure 11.* Euclidean distance of task representations on Cheetah-Vel.

Cheetah-Vel is a relatively simple dense-reward task, most methods can capture a meaningful distance pattern to some extent: tasks with similar target velocities tend to be closer in the learned representation space, while tasks with larger velocity gaps are generally farther apart. MetaSTAR also preserves this locally smooth structure, indicating that its task embeddings are aligned with the underlying velocity semantics.

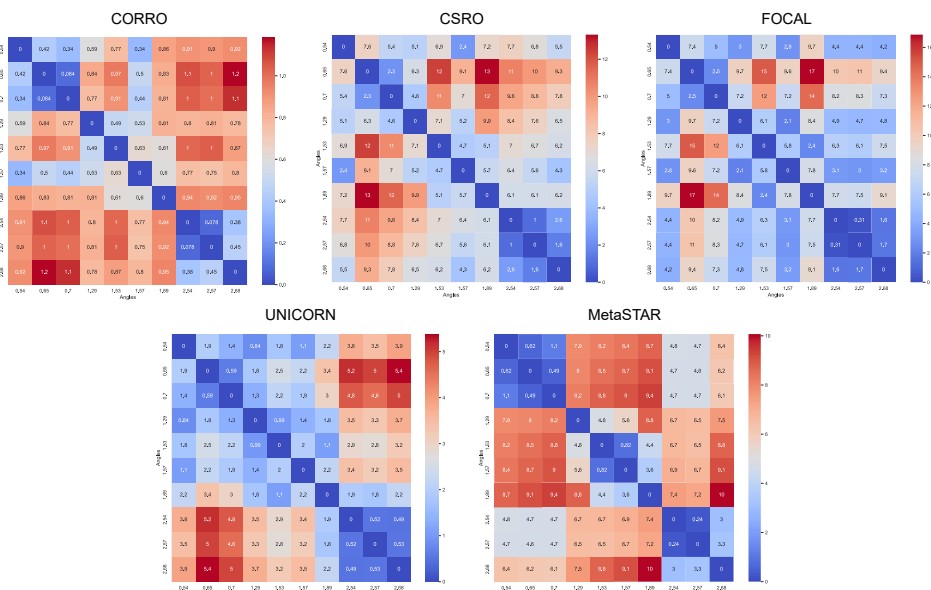

*Figure 12.* Euclidean distance of task representations on Point-Robot-Sparse.

Figure 12 shows the results on Point-Robot-Sparse, where the goal position is $(\cos\theta, \sin\theta)$ and $\theta$ is sampled from $[0.0, \pi]$. MetaSTAR also produces a distance pattern that is broadly consistent with the underlying task geometry: tasks with closer goal positions or similar angular structure tend to have smaller representation distances. This indicates that the representation learned by MetaSTAR captures geometric relationships among sparse-reward tasks, even though reward feedback is limited and task-identifying signals are harder to observe. We also note that angular distance alone does not always fully determine the representation distance in Point-Robot-Sparse. Some tasks with relatively different angles may still appear closer in the learned representation space because their goal positions share similar geometric components, such as similar $\sin\theta$ values.

This behavior is reasonable because the task is defined by the position $(\cos\theta, \sin\theta)$ rather than by the scalar angle alone.

## D.5. Ablation Results

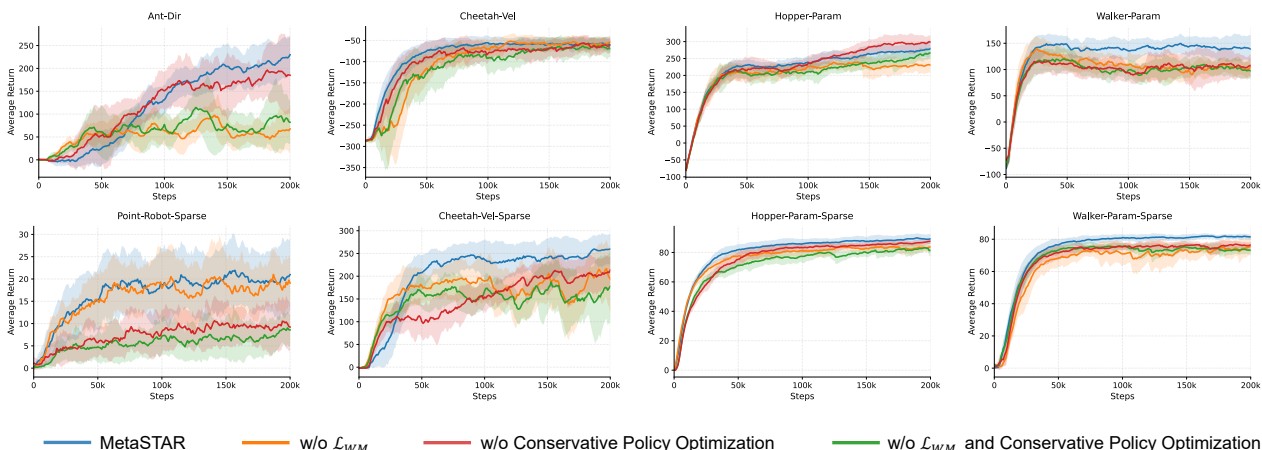

*Figure 13.* Ablation study on online testing across 8 environments, to compare MetaSTAR with methods that without conservative policy optimization and/or $\mathcal{L}_{\mathrm{WM}}$.

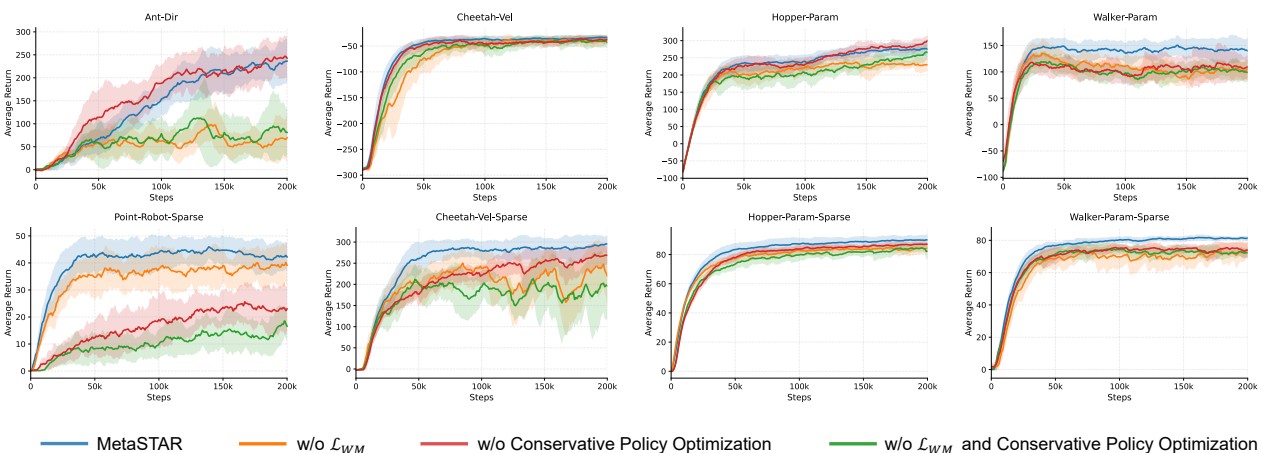

*Figure 14.* Ablation study on offline testing across 8 environments, to compare MetaSTAR with methods that without conservative policy optimization and/or $\mathcal{L}_{\mathrm{WM}}$.

Figure 13 and Figure 14 provide the full online and offline ablation curves across eight environments, complementing the ablation summary in the main text. The ablated variants remove either the world-model objective $\mathcal{L}_{\mathrm{WM}}$, conservative policy optimization, or both components. These curves allow us to examine not only the final performance but also the training dynamics of different variants.

We first observe that removing $\mathcal{L}_{\mathrm{WM}}$ generally weakens performance across most environments. This effect is more visible in tasks where task inference is crucial for adaptation, such as Ant-Dir and sparse-reward environments. Without the world-model dynamics objective, the task representation is mainly shaped by metric-level separation, but receives less direct supervision from transition- and reward-related prediction. Consequently, the learned embedding may be less informative about the underlying task dynamics, which can further affect downstream policy learning.

Removing conservative policy optimization also leads to performance degradation in many settings, especially in online testing. This is consistent with the role of the conservative objective: when imagined rollouts are used for policy improvement, the policy may otherwise exploit unsupported state-action regions or model prediction errors. By penalizing overly optimistic values on policy-induced regions that are insufficiently supported by the offline data, conservative policy optimization

stabilizes imagination-based policy learning.

When both components are removed, the model becomes a simplified metric-learning variant without latent dynamics learning or conservative imagination-based policy optimization. This variant usually suffers the largest degradation, particularly in sparse-reward environments. This suggests that metric-based representation learning alone is not sufficient for robust adaptation when task-identifying reward signals are limited and the policy must generalize from static offline data.

The offline curves in Figure 14 show a similar trend, but the gaps between variants are sometimes smaller than those in online testing. This is expected because offline testing uses pre-collected contexts and therefore avoids part of the context distribution shift introduced by online exploration. In contrast, online testing in Figure 13 better reflects the deployment setting, where both task inference and policy learning are affected by the mismatch between offline data and online interaction.

Finally, the contribution of conservative policy optimization is task-dependent. In some dense-reward environments, such as Hopper-Param, removing conservative policy optimization can slightly improve performance. One possible explanation is that dense rewards and relatively informative offline trajectories reduce the need for additional imagined data, while model-generated rollouts may introduce extra bias. Nevertheless, across the full set of environments, the results support the complementary roles of the two components: $\mathcal{L}_{\mathrm{WM}}$ improves task representation under context distribution shift, while conservative policy optimization stabilizes policy learning under policy distribution shift.

### D.6. Effect of Previous Reward in Observations

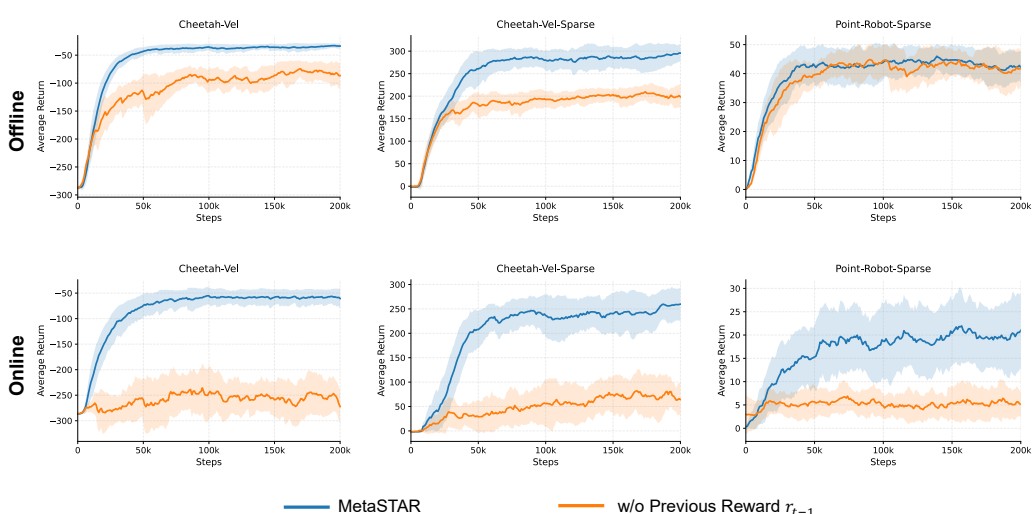

*Figure 15.* Effect of removing the previous reward $r_{t-1}$ from the world-model observation on Cheetah-Vel, Cheetah-Vel-Sparse, and Point-Robot-Sparse.

We additionally ablate the use of the previous reward $r_{t-1}$ in the world-model observation, where the observation is defined as $o_t = [s_t, r_{t-1}]$. Figure 15 compares the full MetaSTAR with a variant that removes $r_{t-1}$ from the observation. The results show that previous reward is an important component of MetaSTAR rather than an incidental implementation detail. On Cheetah-Vel, removing $r_{t-1}$ leads to clear performance degradation in both offline and online testing, with a more pronounced drop in the online setting. On Cheetah-Vel-Sparse, the performance also decreases noticeably after removing $r_{t-1}$, suggesting that previous reward still provides useful task-related cues even when rewards are sparse. On Point-Robot-Sparse, the offline testing gap is relatively small, but online adaptation becomes substantially worse without $r_{t-1}$. These results indicate that $r_{t-1}$ serves as an additional temporal signal for task inference. Although previous rewards are often zero in sparse-reward tasks, their occurrence pattern still carries information about the reward structure and helps the world model associate latent dynamics with task-specific feedback. This signal is particularly useful during online adaptation, where contexts are collected by the agent's exploration policy rather than sampled from the offline behavior data. Without this reward signal, the task representation becomes less informative, leading to weaker adaptation under context distribution shift.

## D.7. Effect of the Contrastive Loss Weight

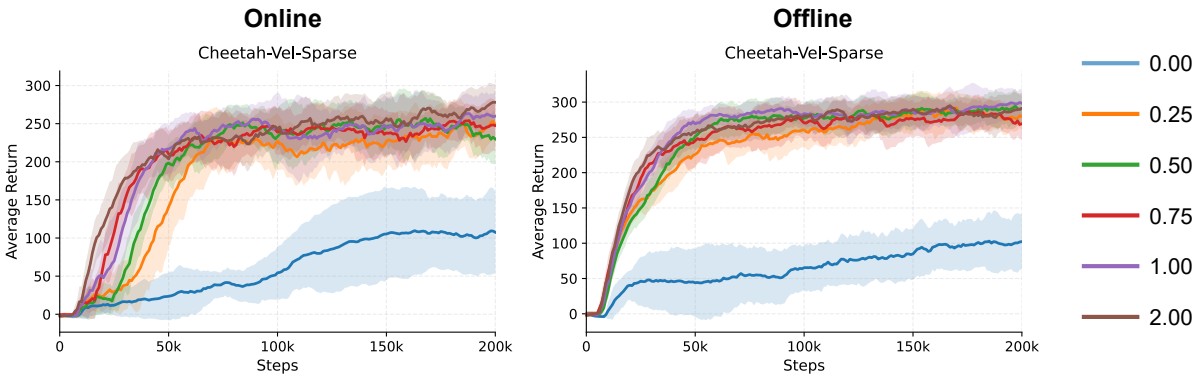

*Figure 16.* Different hyperparameter settings of $\lambda$ on Cheetah-Vel-Sparse.

We conduct a sensitivity analysis on the contrastive loss weight $\lambda$ in Eq. (8). As shown in Figure 16, when $\lambda > 0$, different values mainly affect the convergence speed in the early stage of training, while having limited influence on the final asymptotic performance. This suggests that the world-model objective $\mathcal{L}_{\text{WM}}$ already provides a stable foundation for task representation learning, while the contrastive objective mainly serves as an auxiliary regularizer that improves task discriminability and accelerates representation convergence. Even with a relatively small $\lambda$, MetaSTAR can still extract task-relevant information and support stable policy learning. As $\lambda$ increases, the contrastive constraint can separate embeddings of different tasks more quickly in the early stage, leading to faster convergence. However, once the task representations become stable, its marginal contribution to final performance becomes less pronounced.

## D.8. Effect of the Conservative Penalty Weight

We further analyze the effect of the conservative penalty weight $\beta$ in conservative meta-value evaluation. Figure 17 shows the results on Cheetah-Vel, Cheetah-Vel-Sparse, and Point-Robot-Sparse. On Cheetah-Vel and Cheetah-Vel-Sparse, a large value such as $\beta = 5.0$ noticeably reduces online testing performance, while smaller or moderate values, such as $\beta = 0.5$ and $\beta = 1.0$, perform better. This suggests that overly strong conservatism can suppress policy improvement by penalizing potentially useful imagined state-action regions. In contrast, the performance differences on Point-Robot-Sparse are relatively small, indicating that this task is less sensitive to the conservative penalty strength.

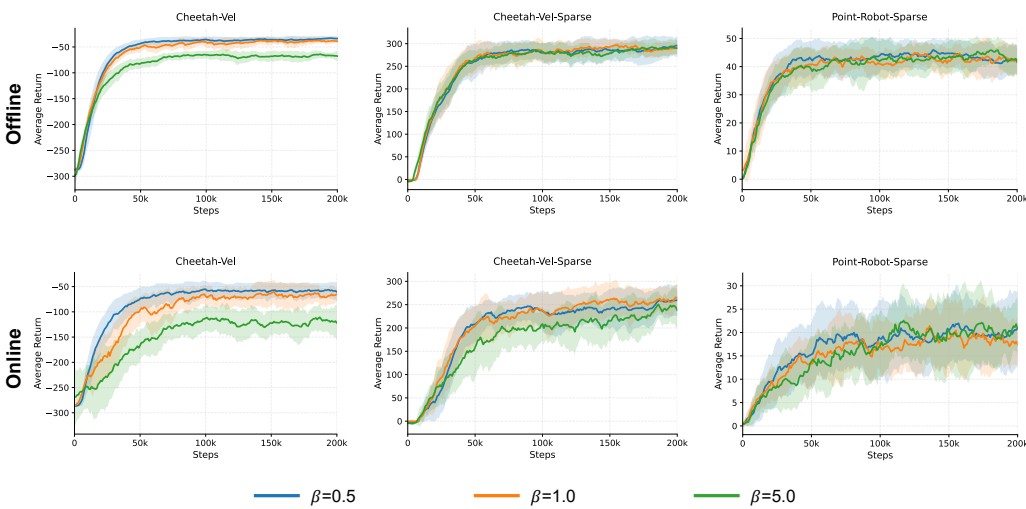

*Figure 17.* Effect of the conservative penalty weight $\beta$ on Cheetah-Vel, Cheetah-Vel-Sparse, and Point-Robot-Sparse.

Mechanistically, $\beta$ controls the strength of pessimistic regularization on unsupported state-action regions, forming a trade-off between exploiting world-model imagination and suppressing model-error propagation. When $\beta$ is small, the policy can

more actively use imagined transitions for optimization, which may improve adaptation but also increases the risk of value overestimation caused by model errors. When $\beta$ is large, the critic imposes stronger penalties on policy-induced regions, which can improve stability but may also lead to overly pessimistic value estimates and limit the upper bound of policy performance.

This trade-off is task-dependent. In dense-reward tasks such as Cheetah-Vel, the offline data often already contain informative trajectories, so excessive conservatism may unnecessarily restrict policy improvement. In Cheetah-Vel-Sparse, imagined rollouts are more useful for extending task-relevant experience, but an overly large $\beta$ can suppress the benefit of such rollouts. For Point-Robot-Sparse, the task structure is relatively simple and easier to identify from limited context, so the policy is less sensitive to the exact value of $\beta$.

### D.9. Effect of the Context Length in Imagination

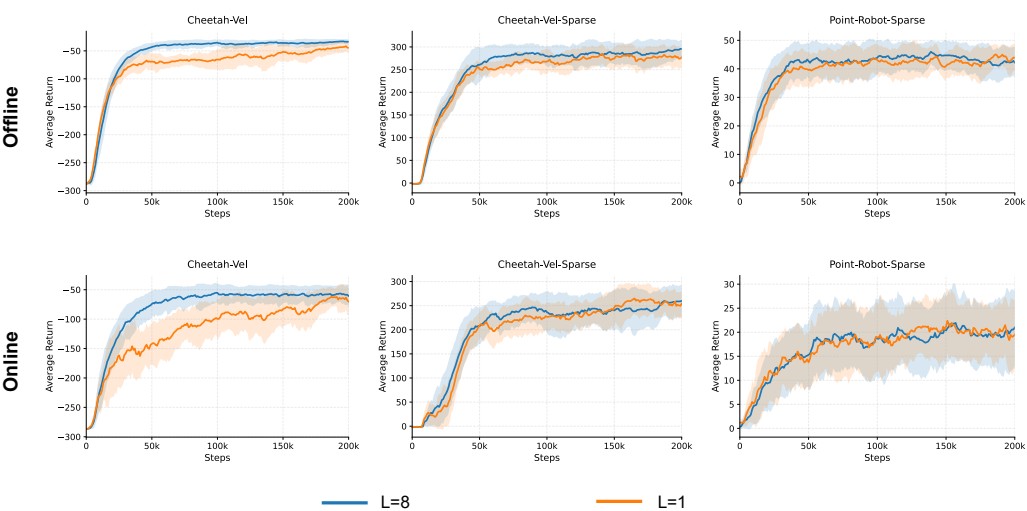

*Figure 18.* Effect of the context length $L$ in contextual imagination on Cheetah-Vel, Cheetah-Vel-Sparse, and Point-Robot-Sparse.

We also investigate the role of the real context length $L$ in contextual imagination by comparing $L = 1$ and $L = 8$. As shown in Figure 18, a longer context prefix generally improves performance. On Cheetah-Vel, $L = 8$ clearly outperforms $L = 1$ in both offline and online testing, with a more pronounced advantage in online testing. On Cheetah-Vel-Sparse, $L = 8$ also performs better overall, although the gap is smaller. On Point-Robot-Sparse, the two settings achieve similar performance, suggesting that this environment is less sensitive to the context length.

These results indicate that the real context prefix plays an important role in world-model-based task inference and imagination. A longer context provides more task-relevant evidence before imagined rollouts are generated, allowing the world model to build a more consistent latent history and reducing ambiguity about the underlying MDP. In contrast, an overly short context, such as $L = 1$, may be insufficient to characterize the current task, especially in environments that require fine-grained task identification. This can lead to less stable task representations and weaker online adaptation.

The effect of $L$ also depends on the environment. In Cheetah-Vel, although single-step transitions already contain some task information, a longer context still improves the consistency of task inference and brings clearer gains during online adaptation. In Cheetah-Vel-Sparse, the reward signal is less informative, so task identification relies more on transition patterns; increasing $L$ remains beneficial, but the improvement is smaller due to the limited task-identifying feedback. In Point-Robot-Sparse, the effective task information contained in short online contexts can be very limited under sparse rewards. Therefore, performance is less determined by context length itself and more by whether the sampled context contains informative transitions, making this environment relatively insensitive to the choice of $L$.

### D.10. Effect of the Imagination Horizon

We then evaluate the influence of the imagination horizon $H$ in contextual imagination. Figure 19 shows the results on Cheetah-Vel, Cheetah-Vel-Sparse, and Point-Robot-Sparse. On Cheetah-Vel, larger values of $H$ slightly slow down early-stage convergence, but different settings eventually reach similar final performance. On Cheetah-Vel-Sparse and

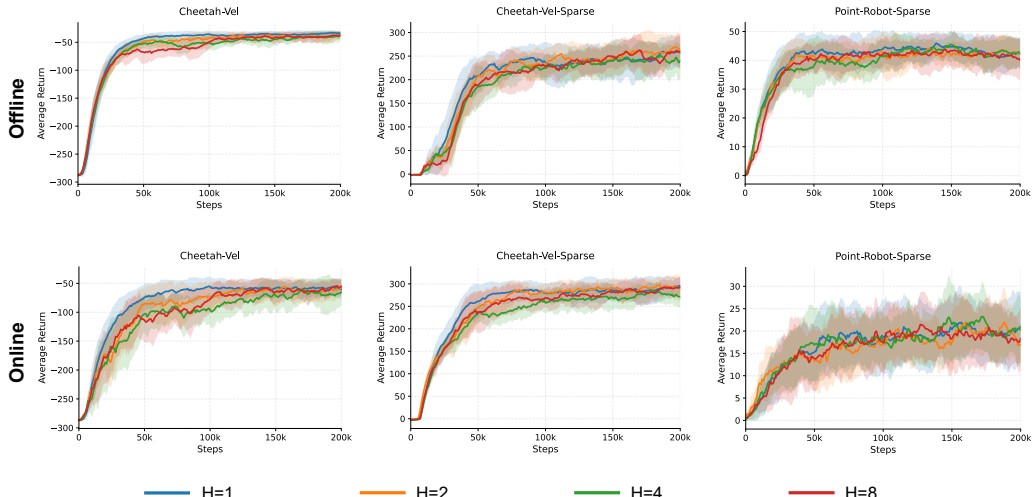

*Figure 19.* Effect of the imagination horizon $H$ on Cheetah-Vel, Cheetah-Vel-Sparse, and Point-Robot-Sparse.

Point-Robot-Sparse, the influence of $H$ is also relatively mild, suggesting that MetaSTAR is not highly sensitive to the imagination horizon within the tested range.

This result reflects the trade-off in imagination-based policy optimization. A longer horizon provides more imagined transitions for policy learning, but it may also introduce additional model bias through accumulated prediction errors. In dense-reward tasks such as Cheetah-Vel, reward signals are frequent and already provide sufficient learning feedback, so a short imagination horizon is usually enough for local model-based augmentation while reducing unnecessary error accumulation. In sparse-reward tasks, reward feedback is much less frequent, and moderately longer context-conditioned rollouts can provide additional imagined transitions that may help policy learning under limited feedback.

Therefore, we can use a short horizon for dense-reward tasks and a moderately longer horizon for sparse-reward tasks. This choice reflects the different role of imagination in the two settings: dense rewards provide frequent learning signals, whereas sparse rewards can benefit from additional context-conditioned imagined transitions. Importantly, the purpose of imagination in MetaSTAR is to conservatively expand the effective support region around real offline contexts, rather than to perform deep trajectory planning as in Dreamer-style agents. Overall, the results indicate that MetaSTAR only requires short-to-moderate horizon imagination for conservative policy learning, while the world model itself still learns task-relevant latent dynamics from full historical contexts.

### D.11. Training Time Analysis

To evaluate the computational cost of MetaSTAR, we report the total offline training time of MetaSTAR and the baseline methods under the same experimental setup. All results are measured on a single RTX 4090 GPU, as shown in Table 3. We note that the absolute wall-clock time may vary across machines due to differences in CPU performance, memory bandwidth, data loading efficiency, and software configuration. Therefore, the table should be interpreted mainly as a relative comparison under the same hardware and implementation setting.

*Table 3.* Training time comparison on a single RTX 4090 GPU, measured in hours.

| Environment | CORRO | CSRO | FOCAL | UNICORN | MetaSTAR |
|---|---|---|---|---|---|
| Ant-Dir | $8.5 \pm 2.4$ | $11.6 \pm 1.7$ | $12.1 \pm 2.6$ | $10.8 \pm 0.7$ | $15.2 \pm 4.4$ |
| Cheetah-Vel | $10.6 \pm 2.5$ | $18.2 \pm 0.3$ | $17.0 \pm 0.9$ | $19.4 \pm 2.4$ | $23.5 \pm 0.5$ |
| Hopper-Param | $9.6 \pm 2.0$ | $16.1 \pm 4.0$ | $13.8 \pm 3.2$ | $17.5 \pm 4.3$ | $21.7 \pm 2.1$ |
| Walker-Param | $8.9 \pm 3.3$ | $13.2 \pm 4.7$ | $12.2 \pm 4.6$ | $14.2 \pm 4.9$ | $17.1 \pm 4.8$ |
| Point-Robot-Sparse | $6.5 \pm 2.3$ | $8.8 \pm 1.8$ | $8.7 \pm 1.7$ | $9.2 \pm 2.7$ | $13.0 \pm 1.7$ |
| Cheetah-Vel-Sparse | $9.4 \pm 2.2$ | $11.5 \pm 1.0$ | $10.9 \pm 0.6$ | $13.3 \pm 0.7$ | $17.5 \pm 1.1$ |
| Hopper-Param-Sparse | $6.2 \pm 1.7$ | $7.5 \pm 0.4$ | $7.3 \pm 0.2$ | $10.3 \pm 1.3$ | $13.8 \pm 0.5$ |
| Walker-Param-Sparse | $9.4 \pm 1.0$ | $16.6 \pm 2.1$ | $16.0 \pm 0.2$ | $17.8 \pm 0.5$ | $20.9 \pm 3.5$ |

Overall, under the same setting, MetaSTAR requires longer offline training time than the model-free baselines. This additional cost mainly comes from two components. First, MetaSTAR trains a stochastic Transformer-based world model, which introduces sequence modeling and latent dynamics prediction beyond standard context-encoder training. Second, during agent learning, MetaSTAR performs contextual imagination on the fly and optimizes the critic using both real and imagined transitions, increasing the cost of each update.

This overhead reflects a trade-off between computational cost and robustness. In offline meta-RL, training is performed once using static datasets, without requiring additional environment interactions. Moreover, at meta-test time, MetaSTAR only needs to infer the task representation and execute the policy through forward passes, so the additional cost is primarily concentrated in offline training rather than deployment. Improving the efficiency of world-model training and imagination-based policy optimization is an important direction for future work.

## E. Context-based Offline Meta-RL Baselines

**FOCAL (Li et al., 2020)**  employs a deterministic context encoder trained via a negative-power distance metric to learn task representations on a bounded embedding space. The framework detaches the metric learning from the Bellman update to mitigate gradient interference during training. Built on the principles of Behavior Regularized Actor-Critic (BRAC), the algorithm addresses value function divergence and distributional shift by regularizing the learned policy against the behavior policy.

**CORRO (Yuan & Lu, 2022)**  is designed to learn robust task representations under behavior-policy mismatch. It features a bi-level task encoder where the first level extracts latent representations from individual transition tuples, rather than full trajectories, while the second level aggregates these representations into a task embedding. The framework employs a contrastive learning objective, formalized as mutual information maximization, to separate task-relevant information from behavior policy features. To optimize this objective, CORRO generates negative pairs using generative modeling via a Conditional VAE (CVAE) or through reward randomization. These task embeddings are then used to condition the policy and value functions within an offline reinforcement learning backbone.

**CSRO (Gao et al., 2023)**  is proposed to address the context-shift issue. During the meta-training phase, the framework employs a max-min mutual information representation learning mechanism. This mechanism uses the CLUB (Contrastive Log-ratio Upper Bound) estimator to minimize the mutual information between task representations and behavior policies while maximizing the mutual information between representations and task-specific information. In the meta-test phase, CSRO utilizes a non-prior context collection strategy, where the agent performs initial random exploration to gather transition data before updating its task posterior, thereby reducing the influence of biased initial task priors. The framework is typically implemented with Behavior Regularized Actor-Critic (BRAC) as the offline backbone to handle value function estimation and policy regularization.

**UNICORN (Li et al., 2024)**  provides a theoretical consolidation of existing methodologies by framing task representation learning as the optimization of mutual information $I(Z; M)$ between the task variable $M$ and its latent representation $Z$. Specifically, the framework demonstrates that prior algorithms—such as FOCAL, CORRO, and CSRO—function as mathematical upper bounds, lower bounds, or linear interpolations of this mutual information objective. UNICORN introduces two primary instantiations: UNICORN-SUP, a supervised variant that utilizes cross-entropy loss for direct estimation, and UNICORN-SS, a self-supervised variant that combines contrastive objectives with generative reconstruction-based regularization. By seeking tighter approximations of $I(Z; M)$, the framework aims to trade off causal and spurious correlations to enhance robustness against context shift and improve generalization in out-of-distribution scenarios.

## F. Environments and Implementation Details

**Ant-Dir.**  The Ant-Dir environment requires controlling an ant robot to move along a target direction on the plane. For each task, a goal direction $\theta \in [0, 2\pi)$ is sampled, which defines a unit direction vector $\mathbf{d} = (\cos\theta, \sin\theta)$. The reward encourages the agent to move along the specified direction by measuring its directional velocity. The state space contains the ant's joint positions, joint velocities, and body posture information, while the action space consists of continuous motor torques applied to the robot joints.

*Table 4.* Episode length and goal settings of environments used.

| Environments | Max steps per episode | Goal type | Goal range | ID tasks Range | OOD tasks Range |
|---|---|---|---|---|---|
| Ant-Dir | 200 | Direction | $[0, 2\pi)$ | $[\frac{\pi}{4}, \frac{7\pi}{4}]$ | $[0, \frac{\pi}{4}) \cup (\frac{7\pi}{4}, 2\pi)$ |
| Point-Robot-Sparse | 60 | Position | $[0, \pi]$ | $[\frac{\pi}{8}, \frac{7\pi}{8}]$ | $[0, \frac{\pi}{8}) \cup (\frac{7\pi}{8}, \pi]$ |
| Cheetah-Vel | 200 | Velocity | $[0, 3]$ | $[0.5, 2.5]$ | $[0, 0.5) \cup (2.5, 3]$ |
| Cheetah-Vel-Sparse | 200 | Velocity | $[0, 3]$ | $[0.5, 2.5]$ | $[0, 0.5) \cup (2.5, 3]$ |
| Hopper-Param | 200 | Body mass and geom friction | $1.5^{[-3,3]}$ | / | / |
| Hopper-Param-Sparse | 200 | | | / | / |
| Walker-Param | 200 | | | / | / |
| Walker-Param-Sparse | 200 | | | / | / |

**Point-Robot-Sparse.** The Point-Robot-Sparse environment requires a point robot to navigate to a target position randomly placed on a unit half-circle. At the beginning of each episode, a goal position is sampled on the half-circle with radius 1. The agent receives a reward only when it enters a radius of $0.2$ around the target. The observation consists of the current position of the robot, and the action controls movement along the $x$ and $y$ directions. Since rewards are provided only near the target, the sparse reward structure makes the task challenging and requires the agent to learn effective navigation with limited feedback.

**Cheetah-Vel.** The Cheetah-Vel environment requires controlling a HalfCheetah robot to reach and maintain a target velocity. For each task, the target velocity $v^\star$ is sampled uniformly from $[0.0, 3.0]$. At each time step, the agent is rewarded according to how well its current velocity matches the target velocity, with a control regularization term. The reward is given by $-|v_t - v^\star| - 0.05\|a_t\|_2^2$, where $v_t = (x_{t+1} - x_t)/\Delta t$, and the second term penalizes large actions. The state space contains the joint positions, joint velocities, and body posture information of the HalfCheetah robot, while the action space consists of continuous motor torques.

**Cheetah-Vel-Sparse.** Cheetah-Vel-Sparse follows the same setting as Cheetah-Vel, except that it uses a sparse reward mechanism. Specifically, the agent receives a reward only when the difference between its current velocity and the target velocity is within a tolerance threshold, with goal radius $0.5$; otherwise, the reward is zero. This sparse feedback makes velocity adaptation more challenging than in the dense-reward version.

**Hopper-Param.** The Hopper-Param environment requires controlling a planar one-legged hopper to remain upright and move forward under randomized dynamics. At the beginning of each task, key physical parameters are perturbed by multiplicative log-scale factors: the body mass and ground friction coefficient are scaled by $1.5^u$, where $u \sim \text{Uniform}[-3, 3]$. The state space consists of the hopper's joint positions, joint velocities, and body posture information, while the action space consists of continuous motor torques. The reward encourages efficient forward locomotion and is given by the forward velocity $(x_{t+1} - x_t)/\Delta t$ plus an alive bonus of $1.0$, minus a small control penalty $10^{-3}\|a_t\|_2^2$.

**Hopper-Param-Sparse.** Hopper-Param-Sparse also controls a planar one-legged hopper under randomized dynamics, but uses a sparse velocity-tracking objective. Let $d_t = |v_t - 1.5|$ denote the deviation between the current velocity and the target velocity $1.5$. The reward is provided only when the agent is close to the target velocity: if $d_t > 0.5$, the reward is 0; otherwise, the reward is $0.8 - d_t$. A control cost $10^{-3}\|a_t\|_2^2$ is subtracted, and no alive bonus is used. This sparse feedback structure provides limited guidance and makes the task more difficult.

**Walker-Param.** The Walker-Param environment controls a planar Walker2D robot to remain upright and move forward under randomized dynamics. At the beginning of each task, the body mass and ground friction coefficient are scaled by $1.5^u$, where $u \sim \text{Uniform}[-3, 3]$. The state space contains the Walker2D robot's joint positions, joint velocities, and body posture information, while the action space consists of continuous motor torques. The reward encourages efficient forward locomotion and is given by the forward velocity $(x_{t+1} - x_t)/\Delta t$ plus an alive bonus of $1.0$, minus a small control penalty $10^{-3}\|a_t\|_2^2$.

**Walker-Param-Sparse.** Walker-Param-Sparse modifies Walker-Param by replacing the dense locomotion reward with a sparse velocity-tracking objective. The reward is determined by the deviation $d_t = |v_t - 1.5|$ between the current velocity and the target velocity $1.5$. If $d_t > 0.5$, the reward is 0; otherwise, the reward is $0.8 - d_t$. In addition, a control cost $10^{-3}\|a_t\|_2^2$ is subtracted to penalize large actions, and no alive bonus is used.

## G. Hyperparameters

*Table 5.* Hyperparameter settings for MetaSTAR in different environments.

| Parameter Name | Ant-Dir | Point-Robot-Sparse | Cheetah-Vel /-Sparse | Hopper-Param /-Sparse | Walker-Param /-Sparse |
|---|---|---|---|---|---|
| Number of Tasks | 40 | 40 | 40 | 40 | 40 |
| Number of Training Tasks | 30 | 30 | 30 | 30 | 30 |
| Number of Evaluation Tasks | 10 | 10 | 10 | 10 | 10 |
| Number of Offline Episodes | 1 | 1 | 1 | 1 | 1 |
| Number of Online Episodes | 2 | 2 | 2 | 2 | 2 |
| Number of Iterations | 1000 | 1000 | 1000 | 1000 | 1000 |
| RL Updates per Iteration | 200 | 200 | 200 | 200 | 200 |
| RL Batch Size | 256 | 256 | 256 | 256 | 256 |
| Task Batch Size | 16 | 16 | 16 | 16 | 16 |
| Context Size | 100 | 30 | 100 | 100 | 100 |
| Policy Layers | [256, 256] | [128, 128] | [256, 256] | [256, 256] | [256, 256] |
| World Model Learning Rate | 0.0003 | 0.0003 | 0.0003 | 0.0003 | 0.0003 |
| Actor Learning Rate | 0.0003 | 0.0003 | 0.0003 | 0.0003 | 0.0003 |
| Critic Learning Rate | 0.0003 | 0.0003 | 0.0003 | 0.0003 | 0.0003 |
| Discount Factor ($\gamma$) | 0.99 | 0.9 | 0.99 | 0.99 | 0.99 |
| Entropy Alpha | 0.2 | 0.01 | 0.2 | 0.2 | 0.2 |
| Transformer Hidden Size | 256 | 128 | 256 | 256 | 256 |
| Number of Transformer Layers | 2 | 2 | 2 | 2 | 2 |
| Number of Transformer Heads | 4 | 4 | 4 | 4 | 4 |
| Contrastive Loss Weight ($\lambda$) | 1.0 | 1.0 | 1.0 | 1.0 | 1.0 |
| Conservative Penalty Weight ($\beta$) | 0.5 | 0.5 | 0.5 | 0.5 | 0.5 |
| Context Length of Imagination | 8 | 8 | 8 | 8 | 8 |
| Imagination Horizon | 1 | 4 | 1/8 | 1/8 | 1/8 |
| Task Embedding Size | 5 | 5 | 5 | 40 | 32 |

