# OpenReview forum: "Behavior-Invariant Task Representation Learning with Transformer-based World Models for Offline Meta-Reinforcement Learning"
_ICML.cc/2026/Conference — ICML 2026 regular_

### Official Review · Reviewer_NFeX · 2026-02-22

**Soundness:** 3
**Presentation:** 2
**Significance:** 3
**Originality:** 2
**Overall Recommendation:** 4
**Confidence:** 3

**Summary:**

The paper tackles offline meta-RL under two major failure modes: context distribution shift induced by heterogeneous behavior policies in offline datasets, and policy shift / model exploitation when deploying/adapting online—both particularly damaging in sparse-reward settings. The authors propose MetaSTAR, which integrates (i) information-theoretic behavior-invariant task representation learning with (ii) a Transformer-based stochastic world model. The core idea is to learn a task latent Z that captures “primary causality” (task-relevant dynamics/reward signals) while suppressing “lesser causality” (behavior-policy artifacts), aligning with prior information-theoretic formulations.
Concretely, the method uses a VAE-style stochastic latent, a Transformer sequence model, and a task query token to aggregate history into a global task embedding z.  It also adds a FOCAL-style metric learning regularizer to encourage discriminative, task-identifiable embeddings.
For policy learning, MetaSTAR performs context-conditioned imagination rollouts (warm-started by real context segments) and trains a conservative critic on a mixture of real and model-generated transitions to mitigate overestimation and exploitation of model errors.
Evaluation spans multiple offline meta-RL benchmarks including Point-Robot-Sparse and several MuJoCo control tasks (Ant-Dir, Cheetah-Vel, Hopper/Walker-Param and sparse variants), with both offline-test and online-test protocols.  MetaSTAR reportedly improves performance and stability especially in sparse-reward and OOD settings, compared against FOCAL/CORRO/CSRO/UNICORN.

**Compliance With Llm Reviewing Policy:**

Affirmed.

**Final Justification:**

The author has addressed my concerns.

**Key Questions For Authors:**

1. Beyond the theoretical TV bound, do you empirically measure invariance directly—e.g., train with multiple behavior policies per task and report $\mathrm{TV}(p(z\mid M,\pi), p(z\mid M,\pi'))$ proxies or embedding divergence statistics? (The paper currently emphasizes qualitative t-SNE evidence.)
2. The author defines $o_t=[s_t, r_{t-1}]$ to help distinguish reward functions. How sensitive is performance to this choice, especially in sparse-reward tasks where r_{t-1} is often zero?
3. Since MetaSTAR uses a model-based pipeline with imagination and extra objectives (world model + FOCAL + conservative critic), do baselines receive comparable tuning and compute budgets? If not, can you provide normalized comparisons under equal wall-clock or equal environment-model forward passes?
4.  In Eq. (8), how are the sampling distributions $\rho$ and $D$ implemented in practice (minibatch construction), and how sensitive is performance to $\beta$, $f$, and the target-network update schedule?

**Limitations:**

The author does not discuss the limitation of the method. Training a stochastic Transformer world model plus additional losses and conservative model-based policy learning is likely resource-intensive and may limit accessibility or scalability to larger observation spaces without careful engineering.

**Strengths And Weaknesses:**

Strength:
1. Using a stochastic world model with Transformer sequence modeling is well-motivated for capturing latent task dynamics and long-horizon dependencies, and the paper explicitly frames why this could “filter” behavior-policy artifacts in offline datasets.
2. The paper explicitly studies OOD splits (ID vs OOD) and emphasizes online-test performance, which is the practically relevant regime for offline meta-RL.

Weakness:
1.  Although the method includes conservative regularization, the training uses a mixture of real and model-generated data and performs rollouts beyond the offline support. It would help to quantify how often imagined transitions drift and how conservative penalties prevent it, rather than relying primarily on aggregate returns.
2. World-model training (VAE + Transformer), task embedding learning, imagination rollouts, and conservative critic training add substantial complexity relative to purely model-free offline meta-RL baselines. The paper would be stronger with explicit compute comparisons (GPU-hours, wall-clock, number of imagined samples) and sensitivity to rollout horizon H, context length L, and mixing factor f.
3. t-SNE clustering is qualitative and can be misleading; stronger tests would involve quantitative invariance metrics (e.g., representation distance across behavior policies for the same task) and downstream identifiability under controlled behavior-policy perturbations.

---

> ### Author Rebuttal · Authors · 2026-03-31
>
> We sincerely thank the reviewer for recognizing the motivation, method design, and the importance of OOD / online-test evaluation. We address the concerns below.
>
> ### Weaknesses
>
> **1. Quantifying imagined drift and conservative penalties.**
>
> We agree that aggregate returns alone are insufficient. Our conservative critic follows the core idea of COMBO: imagined transitions may go beyond offline support, so we penalize Q-values there while anchoring learning on real offline data. To make this more explicit, we added a $\beta$ sensitivity study. The results of **Fig. 2** in [additional experiments](https://anonymous.4open.science/r/MetaSTAR-Codes/README.md) show a clear tradeoff: smaller $\beta$ makes the policy more aggressive and improves exploration/generalization, but it is more vulnerable to model error; larger $\beta$ is safer and more stable, but can under-estimate and suppress performance. On Cheetah-Vel and Cheetah-Vel-Sparse, overly large $\beta=5.0$ hurts performance, while $\beta=0.5$ or $1.0$ works better.
>
> **2. Compute cost and sensitivity to $H/L/f$.**
>
> We agree these should be reported more clearly. We therefore added:
>
> - **GPU time:** wall-clock training time on a single **RTX 4090**. MetaSTAR is generally slower than CORRO/CSRO/FOCAL/UNICORN, mainly due to imagination rollouts and conservative critic training. We will explicitly state this as a limitation. (**Table 1**)
> - **Imagination horizon $H$:** increasing $H$ degrades performance on Cheetah-Vel, while Cheetah-Vel-Sparse and Point-Robot-Sparse are less sensitive, indicating stronger error accumulation in some dense-reward tasks. (**Fig. 3**)
> - **Imagine context length $L$:** we compared $L=1$ and $L=8$. On Cheetah-Vel, $L=8$ is clearly better in both offline and online tests, especially online. On Cheetah-Vel-Sparse, $L=8$ is still better but the gap is smaller; on Point-Robot-Sparse, the difference is minor. This shows that longer real context provides stronger task-identifying evidence and better grounds imagination.  (**Fig. 4**)
> - **Mixing factor $f$:** we follow the COMBO formulation. In practice, each critic batch has size 256, with 240 real and 16 imagined transitions, so $f=\frac{240}{256}=0.9375$. Thus, learning is still dominated by real offline data, while a small amount of imagined data expands support conservatively.
>
> **3. t-SNE is qualitative; stronger invariance evidence is needed.**
>
> We agree. We therefore added **representation distance matrix** analyses on Cheetah-Vel, Cheetah-Vel-Sparse, and Point-Robot-Sparse (**Fig. 5-7**). In the first two, representation distance grows consistently with target velocity difference. In Point-Robot-Sparse, the pattern is also largely consistent with angular difference. There are also cases such as $\theta=0.54$ and $\theta=2.68$ where angular difference is large but representation distance is not, because their sine values are close, indicating similar geometry in the underlying goal space. This suggests that MetaSTAR captures task geometry rather than merely scalar order.
>
> ### Key Questions
>
> **Q1. Do you measure invariance directly beyond the TV bound?**
>
> Yes. The added **representation distance matrix** is a more direct empirical measure than t-SNE. Also, **online test / online adaptation** is itself a functional test of behavior invariance: if the learned representation were dominated by offline behavior-policy-specific action patterns, task inference should fail once the context is collected by the agent’s own exploratory behavior. This is not what we observe.
>
> **Q2. How sensitive is $o_t=[s_t,r_{t-1}]$, especially when $r_{t-1}=0$ in sparse rewards?**
>
> We added an ablation without previous reward. **Fig. 1** shows that removing $r_{t-1}$ causes large degradation on Cheetah-Vel and Cheetah-Vel-Sparse, and also strongly hurts **online adaptation** on Point-Robot-Sparse. Although $r_{t-1}$ is often zero in sparse-reward tasks, it still provides useful reward-related temporal context.
>
> **Q3. Are baselines given comparable tuning / compute budgets?**
> We now report training time for all methods on a single RTX 4090. MetaSTAR is indeed more expensive, and we will explicitly state this as a limitation. We will also phrase our claims more carefully: MetaSTAR’s main advantage is in **sparse-reward** and **OOD online adaptation**, rather than compute efficiency.
>
> **Q4. How are $\rho$ and $d_f$ implemented, and how sensitive are $\beta$ and $f$?**
>
> - $\rho(s,a)$ is the state-action marginal induced by rolling out the current policy in the learned model;
> - $d_f$ is the mixture of real offline and imagined data.
>   In practice, each critic minibatch is built from real and imagined transitions as described above. We have added sensitivity analyses for $\beta$, $H$, and $L$, and will include more implementation details in the appendix.
>
> We thank the reviewer again for the constructive suggestions. We will revise the paper accordingly to improve clarity and empirical support.

---

> > ### Author Rebuttal · Reviewer_NFeX · 2026-04-03
> >
> > Thank you for the response. I'm happy to adjust my score.

---

> > > ### Author Response · Authors · 2026-04-03
> > >
> > > Thank you very much for the positive update. We are glad that our rebuttal has adequately addressed your concerns.
> > >
> > > We sincerely appreciate your careful reading, constructive feedback, and your willingness to adjust the score. Thank you again for your time and support.

---

### Official Review · Reviewer_3nE8 · 2026-03-12

**Soundness:** 3
**Presentation:** 2
**Significance:** 2
**Originality:** 2
**Overall Recommendation:** 4
**Confidence:** 4

**Summary:**

This paper presents a novel offline meta RL framework that overcomes the behaviour policy distribution shift problem using a world model to learn behavior-invariant task representations. They theoretically prove that the world model learns a behaviour-invariant task representation and provide experimental validation of their method.

**Compliance With Llm Reviewing Policy:**

Affirmed.

**Final Justification:**

Clarifications on ICRL baselines, computational costs, train task distribution and seeds were given.

**Key Questions For Authors:**

1. How does MetaSTAR perform compared to other meta offline RL baselines that use transformer backbones like DPT and AD?
2. How does MetaSTAR perform when trained on suboptimal datasets?

**Limitations:**

The constraint of computational costs and the quality requirements of the dataset have not been discussed. Limitations and future work should be addressed in the conclusion.

**Strengths And Weaknesses:**

Strengths:
1. Proposed framework is novel and sound, this is a relevant problem and the proposed solution is elegant.
2. Theoretical validation. The authors theoretically show that extracting the latent task dynamics from the world model helps to improve robustness under behaviour policy shifts.


Weaknesses:
1. No discussion of computational cost.
2. Comparisons to other offline meta RL baselines missing, such as DPT, AD, PEARL. Please expand on using other offline meta RL baselines, including in-context reinforcement learning based ones.
3. In the experiments, please provide more details on the distribution shift between train and test tasks. How much do they differ? Please quantify.
4. No indication of seed number in the experimental section, it is unclear if the results are significant. Please provide more details on the seeds.
5. MetaSTAR underperforms baselines in some environments, discuss this and provide an explanation.
6. The offline datasets were generated by a trained RL agent, that assumption is not realistic in many real-world applications. How does the method perform when using offline datasets of poorer quality, e.g. generated by static baselines?

---

> ### Author Rebuttal · Authors · 2026-03-31
>
> We sincerely thank the reviewer for the positive assessment of our paper. We understand the main concerns to be **computational cost, baseline coverage, train/test shift, random seeds, performance in some environments, and dataset quality**. We will clarify these points in the revision.
>
> ## Weaknesses
>
> **1. Computational cost.**
> We agree that the current version does not discuss training cost sufficiently. In our [additional experiments](https://anonymous.4open.science/r/MetaSTAR-Codes/README.md), we report the **training time and compute resources** of MetaSTAR. **Table 1** shows the training time of each method on a single RTX 4090 across environments. MetaSTAR is usually slower than other baselines, so its gains do come with extra computation. We will explicitly add this limitation. The overhead mainly comes from: (i) **world-model-based imagination rollouts**, and (ii) **conservative critic learning** on imagined data. **Fig. 3** also shows that a larger imagination horizon further increases cost and may accumulate model errors, especially on Cheetah-Vel.
>
> **2. Why not compare with DPT/AD/PEARL?**
> Our main comparisons are with FOCAL, CORRO, CSRO, and UNICORN, because these are the most directly comparable **context-based offline meta-RL** methods under similar settings and evaluation protocols.
> For **PEARL**, prior offline meta-RL works(FOCAL, CORRO and CSRO) have already compared against it and generally found it less competitive in this setting. Thus, outperforming these stronger baselines also indirectly suggests improvement over PEARL.
> For **DPT** and **AD**, they are closer to **in-context RL** than to the **offline meta-RL** setting studied here. Their formulation, context usage, objectives, and evaluation protocols are not fully aligned, which is also why existing offline meta-RL papers usually do not treat them as core baselines.
>
> **3. Train/test distribution shift.**
> We agree that this should be stated more clearly. We consider two settings:
>
> - **ID:** 40 tasks are sampled from the same distribution; 30 are used for meta-training and 10 for testing.
> - **OOD:** in several environments, we split the task space into different train/test sub-regions according to target ranges, and evaluate **online adaptation** on held-out OOD tasks. Ant-Dir, Point-Robot-Sparse, Cheetah-Vel, and Cheetah-Vel-Sparse all include ID/OOD splits, with details in **Appendix Table 3**.
>
> **4. Random seeds.**
> We use **3 random seeds (1, 2, 3)** and report mean and variance. We agree this should be explicitly stated in the experiment section.
>
> **5. Why is MetaSTAR not the best in every environment?**
> We agree this deserves more discussion. The main advantage of MetaSTAR appears in **sparse-reward settings** and in **stable OOD online adaptation**. In some dense-reward tasks, such as Cheetah-Vel, task identification is already relatively easy, so the benefit of behavior-invariant task representation is smaller. In addition, the gain from world models + imagination is more evident when latent task structure is harder to infer and online exploration is more costly. In easier dense-reward tasks, the extra model complexity may bring limited benefit. Fig. 3 supports this: increasing the imagination horizon hurts performance on Cheetah-Vel, suggesting higher sensitivity to world-model prediction error.
>
> **6. Dataset realism and suboptimal data.**
> Our offline datasets are **not** composed only of final trajectories from a converged RL policy. Following standard offline meta-RL practice, they are built from the **replay buffer accumulated throughout training**. Therefore, each task naturally contains data from **multiple proficiency levels**, including near-random, intermediate, and stronger behaviors. Thus, the current datasets are already **mixed-quality**, not purely expert data. This is consistent with prior methods such as FOCAL, CORRO, CSRO, and UNICORN. We agree that evaluating on explicitly lower-quality or purely suboptimal static datasets is an important future direction, and we will add this as a limitation.
>
> ## Key Questions
>
> **Q1. How does MetaSTAR compare with DPT and AD?**
> As noted above, DPT and AD are closer to **in-context RL ** than to the **offline meta-RL** setting considered here. Therefore, a direct main-table comparison would not be fully fair. We will clarify this more explicitly in the revision.
>
> **Q2. How does MetaSTAR perform on suboptimal datasets?**
> Our current datasets are already **mixed-quality**, because they come from replay buffers collected throughout training rather than from a single converged policy. Thus, they include data ranging from near-random to stronger behaviors. That said, evaluating on explicitly controlled **purely suboptimal / static datasets** would be valuable future work.

---

> > ### Author Rebuttal · Reviewer_3nE8 · 2026-04-03
> >
> > Thank you for the response. After getting these clarifications and the additional result on compute resources, I am happy to adjust my score.

---

> > > ### Author Response · Authors · 2026-04-03
> > >
> > > Thank you very much for the positive update. We are glad that our clarifications and the additional experiments results have adequately addressed your concerns.
> > >
> > > We sincerely appreciate your time and your willingness to adjust the score. Thank you again for your thoughtful feedback.

---

### Official Review · Reviewer_vq5n · 2026-03-12

**Soundness:** 3
**Presentation:** 4
**Significance:** 2
**Originality:** 2
**Overall Recommendation:** 4
**Confidence:** 4

**Summary:**

This paper proposes MetaSTAR, an offline meta-RL framework that utilizes a Transformer-based stochastic world model for dynamics modeling to extract behavior-invariant task representations. Furthermore, it introduces conservative value regularization into the contextual imagination rollouts generated by the world model. Experiments on several MuJoCo and PointRobot environments including sparse reward and out-of-distribution settings report superior online adaptation performance compared to multiple offline meta-RL baselines.

**Compliance With Llm Reviewing Policy:**

Affirmed.

**Final Justification:**

The authors have made meaningful efforts by adding cross-policy experiments that partially support the behavior-invariance claim and I'll raise my score to 4.

**Key Questions For Authors:**

1. There is a potential self-consistency issue in Definition 2.1. On one hand, the definition formulates $Z=f(e_{1:t},a_{1:t})$, which explicitly depends on the action sequence, where the action distribution is precisely the primary carrier of behavior policy variance. On the other hand, the text claims that it "discards spurious correlations induced by different behavior-specific action distributions." Could the authors clarify in what precise sense $Z$ is invariant to behavior policy, despite being inferred from actions that are themselves shaped by the behavior policy?
2. In the Section 4.1 experimental setup, the authors state: "The offline dataset consists of the experience replay buffers generated by the independently trained SAC agents." This implies that for each task, the dataset is generated by a single, independently trained SAC agent. There is no explicit indication that multiple policies of varying proficiency levels were used to construct a mixed replay buffer for the *same* task. Given that the core contribution of this paper is the acquisition of "Behavior-Invariant" representations, the current experiments fail to provide compelling evidence for this claim, suggesting a potential over-interpretation of the results. Could the authors address this discrepancy?
3. The paper emphasizes that Contextual Imagination resolves MDP ambiguity. However, original Transformer-based world models inherently perform context inference. This contribution seems close to a repackaging of an existing mechanism with new terminology rather than a novel algorithmic innovation. Could the authors elaborate on the fundamental differences and specific novelties here?

**Limitations:**

yes

**Strengths And Weaknesses:**

**Strengths**

1. It integrates a Transformer-based world model into the offline meta-RL setting
2. The experiments demonstrate good performance under sparse reward settings.
3. The paper is clearly written and easy to follow.

**Weaknesses**

1. Definition 2.1 itself is informal, and its connection to the later information-theoretic formalization is not made sufficiently explicit.
2. The proposed approach largely combines existing components, resulting in limited and incremental novelty.
3. The experimental results fail to provide robust support for the paper's claims regarding "behavior-invariant" representations.

---

> ### Author Rebuttal · Authors · 2026-03-31
>
> We thank the reviewer. The main concerns are Definition 2.1, method novelty, and whether the experiments support the claim of learning a behavior-invariant representation.
>
> ## Weaknesses
>
> **1. Definition 2.1 and the theory.**
>
> We agree that Definition 2.1 is not formal enough and that its link to the later information-theoretic analysis should be clearer. In the revision, we will rewrite it more rigorously and align it explicitly with Theorems 3.1-3.4.
>
> Our point is not that the task representation is independent of actions. Rather, although actions are influenced by the behavior policy, the learned representation should focus on **task-related latent dynamics** conditioned on latent history and actions, instead of spuriously encoding behavior-specific action statistics. Thus, “behavior-invariant” does **not** mean action-free; it means **not spuriously tied to behavior-policy-specific action distributions**.
>
> This is supported by our online protocol. At test time, the context is collected through **online interaction**, not drawn directly from the offline behavior-policy distribution. If the representation mainly memorized training-time action patterns, online adaptation should deteriorate. However, MetaSTAR still performs in **online testing** and **OOD adaptation**, suggesting it captures stable task-related information.
>
> **2. Is the method only a combination of existing modules?**
>
> Our contribution is not that world models, contrastive regularization, or conservative value learning are individually new. Rather, we **unify them for offline meta-RL** to address two coupled bottlenecks: **context distribution shift**, where heterogeneous behavior policies can make task representations overfit behavior-specific patterns, and **policy distribution shift**, where imagined rollouts may exploit world-model errors and overestimate unsupported regions.
>
> Accordingly, we use a **Transformer-based stochastic world model** to learn latent task dynamics, a **task token + task embedding projector** to make the model also serve as a task identifier, and **contextual imagination + conservative critic learning** to expand support while controlling overestimation.
>
> **3. Do the experiments sufficiently support the claim?**
>
> Our evidence does not rely only on visualization. A key argument comes from the **online test** and **online adaptation** results. The representation is learned from offline data, whereas the test context is generated by online exploration. If the representation merely memorized behavior-specific action patterns, task inference should become unstable online and performance should drop. Instead, MetaSTAR remains stable and strong relative to the baselines.
>
> We further provide **Euclidean-distance analyses** in Cheetah-Vel, Cheetah-Vel-Sparse, and Point-Robot-Sparse. As shown in the supplementary results ([Additional Experiments](https://anonymous.4open.science/r/MetaSTAR-Codes/README.md), **Fig. 5-7**), MetaSTAR preserves meaningful task geometry: similar target velocities or goal locations map to nearby representations, while dissimilar tasks are farther apart. This suggests task semantics and local continuity, rather than arbitrary clustering or simple memorization of behavior policies.
>
> ## Key Questions
>
> **Q1. If $Z=f(e_{1:t}, a_{1:t})$ depends on actions, why call it behavior-invariant?**
>
> Our claim is **not** that $Z$ ignores actions. Actions are necessary because transition dynamics are conditioned on actions. The key distinction is between **using action information** and **being dominated by behavior-policy bias**. The representation may use actions for task identification, but it should not over-encode spurious action statistics induced by a particular behavior policy.
>
> **Q2. Is each task dataset produced by only one SAC agent, making “behavior-invariant” overstated?**
>
> No. The dataset is **not** composed only of final trajectories from a single converged policy. As in standard offline meta-RL practice, we use the **replay buffer accumulated during training**, which naturally contains early near-random behavior, intermediate policies, and later competent policies. Thus, each task dataset already includes mixed behavior distributions with different proficiency levels.
>
> **Q3. Is “contextual imagination” simply a repackaging of the fact that a Transformer world model already uses context?**
>
> We agree that any sequence-based world model uses history, and this alone is not new. Our point is more specific: in MetaSTAR, **contextual imagination** means that before imagination rollouts, we first use real historical context to warm up task identification, and then roll out from a task-consistent latent history. This is important in offline meta-RL because of **MDP ambiguity**. **Fig. 4** shows the influence of imagination context length.

---

> > ### Author Rebuttal · Reviewer_vq5n · 2026-04-01
> >
> > Thank you for the detailed response. After careful consideration, I find that two core concerns remain insufficiently addressed.
> >
> > **Regarding the empirical validation of behavior invariance.** Your rebuttal clarifies that the replay buffer accumulated during SAC training contains behaviors from different training stages, but I do not think this by itself constitutes adequate evidence for behavior invariance. Policies sampled from a single SAC training trajectory share the same architecture, objective, and optimization path—this is fundamentally different from truly distinct behavior policies (e.g., SAC, TD3, random).
> >
> > **Distinguishing task discriminability from behavior invariance.** The newly added distance matrices (Figs. 5–7) demonstrate task discriminability, not behavior invariance—these are distinct properties. A behavior-sensitive encoder with sufficient capacity can equally achieve inter-task separability. Your argument that strong online performance implies behavior-invariant representations is also inconclusive, as this success may instead stem from the world model's generalization ability or conservative value learning. Notably, the ablation study (Fig. 10) shows substantial performance degradation upon removing conservative policy optimization, suggesting that online performance is materially influenced by this component rather than representation quality alone. Strong online performance is consistent with behavior-invariant representations, but does not by itself identify that property as the operative cause.

---

> > > ### Author Response · Authors · 2026-04-03
> > >
> > > Thank you for the follow-up. We agree that the following points need clearer explanation:
> > >
> > > ## On the empirical validation of behavior invariance
> > >
> > > In our view, **online testing provides a natural and effective functional evaluation of behavior invariance in offline meta-RL**. The goal of learning behavior-invariant task representations is to mitigate **context shift**, as formalized in [CSRO](https://arxiv.org/pdf/2311.03695), i.e., the mismatch between training contexts from the **behavior policy** and test contexts from the **exploration policy**. If a representation were overly tied to training-time behavior patterns, then changing the context-collection policy at test time would disturb task inference and hurt online adaptation. Thus, stable **online test** performance remains an important indicator of robustness to context shift. This is also the basic evaluation used by **CSRO**.
> > >
> > > We also claim that even within one algorithm such as **SAC** or **TD3**, the policy is not static: it evolves during training, so the replay buffer is generated by a **family of policies**, not one fixed policy. Definitely, we fully agree that this is still weaker than comparing distinct policy classes such as **SAC, TD3, and random**. Therefore, we added stricter **cross-policy** experiments (**Fig. 8–16** in the [Additional Experiments](https://anonymous.4open.science/r/MetaSTAR-Codes/README.md); TD3 implementation: [TD3 code](https://anonymous.4open.science/r/MetaSTAR-Codes/algorithms/td3.py)). The new results show:
> > >
> > > - In the **dense-reward** task Cheetah-Vel (**Fig. 8-10**), almost all methods cluster same-task representations reasonably well under SAC, TD3, and random contexts. This suggests task identification is relatively easy here, so behavior-policy change is less challenging.
> > > - In the **sparse-reward** task Cheetah-Vel-Sparse (**Fig. 11-13**), MetaSTAR’s advantage becomes much clearer. Both t-SNE and distance matrices show that MetaSTAR preserves task structure better under TD3 and random contexts, whereas baselines are more distorted.
> > > - In the **extremely sparse** task Point-Robot-Sparse (**Fig. 14-16**): MetaSTAR learns much cleaner representations under SAC and TD3 contexts. Under random-policy contexts, however, nearly all methods degrade. We believe this is reasonable, because in such an extremely sparse setting, random trajectories do not contain enough **task-identifying evidence**, so all methods struggle.
> > >
> > > ## Task discriminability is not behavior invariance
> > >
> > > We fully agree. Therefore, we do **not** claim that distance-matrix separability alone proves behavior invariance.
> > >
> > > What we want to emphasize is: **if the geometry of the learned task representation remains aligned with the true structure of the task goals, this suggests that the representation focuses more on the task itself rather than on superficial patterns induced by a specific behavior policy.** Thus, the added cross-policy experiments do not merely show that MetaSTAR separates tasks; they show that when the context-collection policy changes from SAC to TD3 or random, the relation between learned task structure and true task-goal structure remains more stable for MetaSTAR.
> > >
> > > ## Clarification on Fig. 10
> > >
> > > We also agree that strong online performance cannot be attributed to representation quality alone; conservative policy optimization is also important. Our point is that MetaSTAR’s performance comes from the **combination** of:
> > >
> > > 1. **world-model-based task representation learning**, which supports more robust task inference under context shift;
> > > 2. **conservative policy optimization**, which stabilizes learning under imagined rollouts and helps exploit the world model safely.
> > >
> > > From Fig. 10, when $L_{WM}$ is removed, performance drops in most tasks, especially **Ant-Dir**, indicating that world-model-based representation learning contributes substantially. In **sparse-reward** tasks, removing conservative policy optimization causes much larger degradation because offline dataset often contains only **suboptimal trajectories**. In that case, imagination expands support and helps search for better trajectories, while conservative optimization prevents the policy from exploiting model errors. These two parts are complementary.
> > >
> > > For **dense-reward** tasks, the problem is relatively easier, and the offline dataset often already contains good or near-optimal trajectories. Therefore, the additional gain from conservative policy optimization is less pronounced. This also explains the phenomenon where performance may improve after removing certain modules in some simple tasks.
> > >
> > > Therefore, we do **not** claim that MetaSTAR’s online success is caused solely by behavior-invariant representation learning. Rather, our claim is that outstanding online adaptation comes from the **combination** of world-model-based representation learning and conservative policy optimization.
> > >
> > > Again, we sincerely thank the reviewer for these helpful follow-up questions.

---

### Official Review · Reviewer_R2pH · 2026-03-13

**Soundness:** 3
**Presentation:** 2
**Significance:** 3
**Originality:** 3
**Overall Recommendation:** 5
**Confidence:** 3

**Summary:**

The MetaSTAR framework addresses offline meta-reinforcement learning. The main goal is to achieve robust generalization to previously unseen tasks within the same environment.

To achieve this, the authors employ a Transformer-based world model trained on multiple offline datasets from differing behavioral policies. The paper shows that the process of learning a world model, i.e. next-state and reward prediction based on previous states and action, simultaneously maximizes 'primary' causality. By defining task-specific variables based on the latent history, this can be used to find causal representations of the tasks. The representations latent space is made well-structured via a contrastive loss. With a conservative Value-function approximation and an actor that is additionally dependent on the task-specific representations, the resulting agent is able to adapt to OOD tasks.

The paper provides a theoretical validation of this property as well as empirical evidence of generalization to in-distribution and out-of-distribution tasks.

**Compliance With Llm Reviewing Policy:**

Affirmed.

**Final Justification:**

While I think that the manuscript can still be improved, the authors have convinced me of the quality and importance of their work. However, I am not too familar with transformer-based world models.

**Key Questions For Authors:**

1. Is it correct that the goal is that a new task will produce a task-vector that is close to/an interpolation of the learned task vectors, s.t. the actoris biased to approach the new task similarly to the learned ones?
2. Is the lesser causality minimized in any way? (iirc this is necessary for generalization under causal decomposition)
3. Doesn't the conservative critic (lowering imagined values) lead to less exploration of 'unsafe' states? Why is that desired? Does that impact the exploration capabilities of the agent negatively?

**Limitations:**

There is no discussion of the limitations. This could include environmental changes, more explorative tasks or unsufficient coverage of the offline data.

**Strengths And Weaknesses:**

## Strengths
+ The paper introduces a novel and promising method for learning task-specific representations.
+ The proposed approach successfully achieves state-of-the-art performance in the domain of offline meta-learning.
+ The authors thoughtfully base and validate their architecture choices using rigorous theoretical and causal proofs.
+ All major claims made throughout the paper are well supported by strong empirical evidence or formal proofs.

## Weaknesses
- The ablation study is currently quite limited; for instance, it would be highly interesting to test the effects of removing the actor's dependency on the latent variable z, as well as removing the contrastive loss entirely.
- The experimental section is missing a crucial comparison to a standard model-based offline reinforcement learning baseline, such as DreamerV4.
- Furthermore, the experiments lack a comparison to existing offline meta-RL world models, like MAMBA [1].
- The exact problem setting could be defined much more clearly; while the text implies the method handles changing world dynamics, the actual experiments suggest it only addresses simple goal changes.
- In the method section, the model is introduced without adequately highlighting which specific components are novel or differ from established world models like Dreamer. It should be made clearer that the main additions are primarily the task embedding projector, the contrastive loss, the conservative critic, and the task-dependent actor.
- There is a missing analysis regarding the t-SNE visualization of tasks that the model was not explicitly trained on, as it would be valuable to see if these unseen tasks map intuitively between similar, known tasks in the latent space.
- The paper completely lacks a dedicated discussion of its limitations.
- Finally, the source code is not available at the time of review, which hinders reproducibility.

---

> ### Author Rebuttal · Authors · 2026-03-31
>
> Thank you for the careful review and positive assessment. Below we address the main concerns.
>
> ## Weaknesses
>
> **1. Limited ablations.**
>
> In context-based meta-RL, the task embedding $z$ is part of the latent state and is essential for fast adaptation; removing $z$ from the actor would collapse the policy toward a shared-policy setting. For the contrastive term, our additional experiments already show that **removing the contrastive loss ($\lambda=0$)** weakens task separability in latent space, although the world model still predicts dynamics. See **Fig. 11**.
>
> **2. Missing DreamerV4 baseline.**
>
> DreamerV4 is a general model-based RL method, not an **offline meta-RL** method. Our setting is **context-based offline meta-RL**, where task inference, context shift, and policy shift are central. Thus, direct comparison is not fully fair. We therefore compare with representative **offline meta-RL** baselines: FOCAL, CORRO, CSRO, and UNICORN. We will clarify this in the revision.
>
> **3. Missing MAMBA baseline.**
>
> MAMBA is also not an offline meta-RL baseline, but a related meta-RL world-model method with a different training/adaptation regime. Existing offline meta-RL works likewise do not use it as a main baseline. We agree it should be discussed more clearly in related work.
>
> **4. Problem setting is unclear.**
>
> Meta-RL studies adaptation within a **task distribution sharing the same state/action spaces**, while tasks differ in dynamics $T_i$ and rewards $R_i$. Our benchmarks include both **goal-varying tasks** (Cheetah-Vel, Ant-Dir, Point-Robot-Sparse) and **dynamics-varying tasks** (Hopper-Param, Walker-Param). We will make this clearer.
>
> **5. Innovation is not highlighted enough.**
>
> Our key contributions are:
> (i) introducing a **Transformer-based stochastic world model** into offline meta-RL task representation learning, and linking latent dynamics learning to **primary causality** and behavior-invariant representation;
> (ii) combining **contextual imagination** with a **conservative critic** to enlarge support while controlling overestimation from unreliable imagined data.
>
> **6. Limited analysis of unseen-task t-SNE.**
>
> We agree visualization alone is insufficient. We therefore added **Euclidean distance matrix analyses** on unseen tasks in Cheetah-Vel, Cheetah-Vel-Sparse, and Point-Robot-Sparse. As shown in **Fig. 5-7** of [Additional Experiments](https://anonymous.4open.science/r/MetaSTAR-Codes/README.md), MetaSTAR preserves semantic structure: nearby tasks in velocity/goal space are also nearby in latent space, while more distinct tasks are farther apart. In Point-Robot-Sparse, cases such as $\theta=0.54$ and $\theta=2.68$ remain close because the true goal is $(\cos\theta,\sin\theta)$ and $\sin(0.54)\approx\sin(2.68)$.
>
> **7. Missing limitations.**
>
> We will explicitly add two limitations:
> (i) extra training cost from world-model learning and imagination rollouts;
> (ii) sensitivity to model error accumulation at larger imagination horizons. As shown in **Fig. 3** of  [Additional Experiments](https://anonymous.4open.science/r/MetaSTAR-Codes/README.md), performance drops on Cheetah-Vel when the horizon increases, while **Cheetah-Vel-Sparse** and **Point-Robot-Sparse** are less sensitive.
>
> **8. Code availability.**
>
> The submitted code link is accessible on our side; if it could not be opened during review, this may be due to network or access restrictions.
>
> ## Key Questions
>
> **Q1. Will a new task map near learned task vectors?**
>
> Broadly yes, but more precisely, unseen tasks should be embedded according to **similarity in dynamics and reward structure**, not by mechanical interpolation. The task-conditioned actor $\pi_\theta(a\mid s,z)$ then exploits this structured latent space for fast adaptation.
>
> **Q2. Is lesser causality explicitly minimized?**
>
> Not as an independent loss. Following the UNICORN view, we suppress behavior-specific correlations **implicitly** through world-model latent dynamics learning. Our Theorems 3.3/3.4 show that when the dynamics KL is well calibrated, dependence on behavior policy is bounded, yielding behavior-policy robustness. Thus our mechanism promotes **primary causality**, which indirectly suppresses lesser causality.
>
> **Q3. Does the conservative critic reduce exploration of unsafe imagined states?**
>
> Yes, and this is necessary in offline meta-RL. The main risk is severe **value overestimation outside the support** due to model error. The conservative critic discourages exploiting unreliable imagined regions, improving stability and OOD generalization, though too much conservatism can hurt final performance. Our added **Fig. 2** shows this trade-off: smaller $\beta$ is more aggressive but less robust, while larger $\beta$ is safer but may under-estimate values.
>
> Thank you again for the constructive suggestions. We will revise the paper to clarify the setting, sharpen the novelty, and strengthen the analysis.

---

> > ### Author Rebuttal · Reviewer_R2pH · 2026-04-03
> >
> > My concerns have been addressed. This field is not my primary field of expertise, and I think that the manuscript can still be improved, by incorporating the changes done through this rebuttal, and by increasing e.g. the Figures visual aspects. However, this is minor, so I am happy to adjust my score.

---

> > > ### Author Response · Authors · 2026-04-03
> > >
> > > We sincerely thank the reviewer for the positive update and for taking the time to carefully consider our rebuttal. We are very glad that our clarifications have adequately addressed your concerns.
> > >
> > > We also appreciate your helpful suggestions regarding the presentation quality, including improving the visual aspects of the figures. We agree that these changes would further improve the manuscript, and we will carefully incorporate them, together with the other revisions discussed in the rebuttal, in the final version.
> > >
> > > Thank you again for your thoughtful feedback and support.

---

### Decision · Program_Chairs · 2026-04-30

**Decision:**

Accept (regular)

**Comment:**

This paper studies offline meta-reinforcement learning under context and policy distribution shifts, particularly in sparse-reward settings. The authors study an important concept: how to learn task representations that remain robust to behavior-policy-induced bias while enabling reliable online adaptation.

Reviewers agreed that the paper is technically solid and well-motivated. The proposed MetaSTAR framework integrates a Transformer-based stochastic world model with information-theoretic task representation learning and conservative policy optimization. This combination is supported by a coherent theoretical perspective linking latent dynamics modeling to behavior-invariant representations, and the paper provides formal analysis to justify this connection. Empirically, the method demonstrates consistent improvements over prior offline meta-RL approaches across multiple benchmarks, including challenging sparse-reward and out-of-distribution settings. The results show clear gains in both online adaptation and OOD generalization, which are the practically relevant regimes.

At the same time, reviewers noted several limitations. The overall method introduces substantial complexity and computational overhead compared to existing baselines. Some reviewers have raised concerns about the methodological novelty as the approach combines several known components rather than introducing a fundamentally new algorithmic paradigm. There were also concerns about the clarity of certain definitions and the strength of empirical evidence specifically isolating behavior-invariant representation learning. In addition, the initial version lacked some ablations and baseline comparisons, and the discussion of limitations could be more complete.

These concerns were discussed during the rebuttal phase, and the authors provided additional analyses, including cross-policy evaluations, sensitivity studies, and clarifications of the problem setting and theoretical claims. These responses addressed most of the reviewers’ concerns and strengthened confidence in the validity of the approach.

Overall, the paper presents a well-founded and practically relevant contribution, with strong empirical performance and a coherent theoretical perspective. Despite some limitations in novelty framing and experimental completeness, the work advances the state of the art in offline meta-reinforcement learning and provides a useful direction for future research. My recommendation is to accept.